# Conformational regulation and target-myristoyl switch of calcineurin B homologous protein 3

Florian Becker[1,2†], Simon Fuchs[1†], Lukas Refisch[3,4], Friedel Drepper[5], Wolfgang Bildl[6], Uwe Schulte[6,7,8], Shuo Liang[1], Jonas Immanuel Heinicke[1], Sierra C Hansen[1], Clemens Kreutz[3,7], Bettina Warscheid[5,7,9], Bernd Fakler[6,7,8], Evgeny V Mymrikov[1,7]*, Carola Hunte[1,7,8]*

[1]Institute of Biochemistry and Molecular Biology, ZBMZ, Faculty of Medicine, University of Freiburg, Freiburg, Germany; [2]Faculty of Biology, University of Freiburg, Freiburg, Germany; [3]Institute of Medical Biometry and Statistics (IMBI), Faculty of Medicine and Medical Center, University of Freiburg, Freiburg, Germany; [4]Institute of Physics, University of Freiburg, Freiburg, Germany; [5]Biochemistry and Functional Proteomics, Institute of Biology II, Faculty of Biology, University of Freiburg, Freiburg, Germany; [6]Institute of Physiology, Faculty of Medicine, University of Freiburg, Freiburg, Germany; [7]CIBSS - Centre for Integrative Biological Signalling Studies, University of Freiburg, Freiburg, Germany; [8]BIOSS Centre for Biological Signalling Studies, University of Freiburg, Freiburg, Germany; [9]Department of Biochemistry, Theodor-Boveri-Institute, University of Würzburg, Würzburg, Germany

*For correspondence:
evgeny.mymrikov@biochemie.
uni-freiburg.de (EVM);
carola.hunte@biochemie.uni-
freiburg.de (CH)

†These authors contributed equally to this work

**Abstract** Calcineurin B homologous protein 3 (CHP3) is an EF-hand $Ca^{2+}$-binding protein involved in regulation of cancerogenesis, cardiac hypertrophy, and neuronal development through interactions with sodium/proton exchangers (NHEs) and signalling proteins. While the importance of $Ca^{2+}$ binding and myristoylation for CHP3 function has been recognized, the underlying molecular mechanism remained elusive. In this study, we demonstrate that $Ca^{2+}$ binding and myristoylation independently affect the conformation and functions of human CHP3. $Ca^{2+}$ binding increased local flexibility and hydrophobicity of CHP3 indicative of an open conformation. The $Ca^{2+}$-bound CHP3 exhibited a higher affinity for NHE1 and associated stronger with lipid membranes compared to the $Mg^{2+}$-bound CHP3, which adopted a closed conformation. Myristoylation enhanced the local flexibility of CHP3 and decreased its affinity to NHE1 independently of the bound ion, but did not affect its binding to lipid membranes. The data exclude the proposed $Ca^{2+}$-myristoyl switch for CHP3. Instead, a $Ca^{2+}$-independent exposure of the myristoyl moiety is induced by binding of the target peptide to CHP3 enhancing its association to lipid membranes. We name this novel regulatory mechanism 'target-myristoyl switch'. Collectively, the interplay of $Ca^{2+}$ binding, myristoylation, and target binding allows for a context-specific regulation of CHP3 functions.

## Editor's evaluation

In this work, the authors provide important mechanistic insights into how the intracellular effector protein Calcineurin B homologous protein 3 (CHP3) can be regulated in a calcium-independent manner to expose its lipid membrane binding site. Compelling evidence demonstrates a binding partner protein (NHE1) triggers a conformation change and exposure of the myristoyl group in CHP3 resulting in membrane association. This provides mechanistic insight into the signalling

mechanisms achieved by CHP3 in a target-binding dependent manner, which will be of broad scientific interest.

## Introduction

The calcineurin B homologous protein 3 (CHP3, tescalcin) belongs to the EF-hand $Ca^{2+}$-binding protein (EFCaBP) family (*Kolobynina et al., 2016*) and is closely related to calcineurin B homologous proteins CHP1 and CHP2 (*Di Sole et al., 2012*). CHPs interact with several isoforms of sodium/proton exchangers (NHEs), and this interaction is required for the localization and function of NHE transporters on the plasma membrane (*Pedersen and Counillon, 2019*). Besides NHEs, CHP3 interacts with calcineurin A (*Gutierrez-Ford et al., 2003*), subunit 4 of COP9 signalosome (*Levay and Slepak, 2014*) and glycogen synthase kinase 3 (GSK3) (*Kobayashi et al., 2015*). Recent studies revealed that the CHP3 expression level correlates with the progression, metastasis, and invasiveness of gastric, renal, and colorectal cancers (*Kang et al., 2016*; *Kim et al., 2019*; *Lee et al., 2018*; *Luo et al., 2019*). In addition, genome-wide association studies in combination with neuroimaging identified CHP3 as a key regulator of neurogenesis for hippocampal volume formation (*Dannlowski et al., 2015*; *Horgusluoglu-Moloch et al., 2019*; *Stein et al., 2012*). Via regulation of GSK3 and calcineurin activities, it seems to counteract cardiac hypertrophy (*Kobayashi et al., 2015*; *Viereck et al., 2020*). Thus, CHP3 is an emerging important player in cellular $Ca^{2+}$ signalling networks involved in the regulation of cell proliferation and development in different tissues. The underlying molecular mechanisms are not well understood.

All three CHPs were shown to undergo conformational changes upon $Ca^{2+}$ binding (*Gutierrez-Ford et al., 2003*; *Liang et al., 2020*). Notably, CHP3 has a lower affinity for $Ca^{2+}$ (0.8 µM) (*Gutierrez-Ford et al., 2003*) in comparison to CHP1 and CHP2 ($K_D$ values below 100 nM) (*Li et al., 2011*; *Pang et al., 2004*). Further, $Ca^{2+}$-binding affinities of CHP1 and CHP2 strongly increased (45- and 42-fold, respectively) upon binding of the NHE1 target peptide (*Li et al., 2011*; *Pang et al., 2004*). This modulation is a common feature in EFCaBPs (*Gifford et al., 2007*) and should also apply to the CHP3 isoform. CHP3 may thus respond to $Ca^{2+}$ signals with conformational changes when the intracellular $Ca^{2+}$ concentration elevates from 100 nM to 1 µM or higher (*Roderick and Cook, 2008*). $Ca^{2+}$-induced conformational changes are characteristic for $Ca^{2+}$ sensor proteins such as calmodulin (CaM) and calcineurin B (*Creamer, 2020*; *Nelson and Chazin, 1998*). At the resting $Ca^{2+}$ level, they adopt a closed conformation, in which the hydrophobic target-binding pocket is occluded. Upon a rise of the intracellular $Ca^{2+}$ concentration, $Ca^{2+}$ binding to EF-hand(s) causes an opening of this pocket providing the structural basis for the transmission of $Ca^{2+}$ signals (*Nelson and Chazin, 1998*). This opening often triggers the binding of EFCaBPs to their target proteins (*Burgoyne et al., 2019*). In addition, $Ca^{2+}$-induced conformational changes of EFCaBPs such as calcineurin B or guanylyl cyclase activating proteins (GCAPs) stably associated with target proteins can affect the function of the latter (*Creamer, 2020*; *Lim et al., 2014*). Recently, we demonstrated an increase of CHP3 hydrophobicity upon $Ca^{2+}$ binding that most likely resulted from the opening of the hydrophobic target-binding pocket (*Liang et al., 2020*). CHP3 binds also $Mg^{2+}$ with a low affinity ($K_D$ = 73 µM) in the absence of $Ca^{2+}$. The presence of 1 mM $Mg^{2+}$ reduces the affinity for $Ca^{2+}$ from 0.8 to 3.5 µM, indicating a direct competition between $Ca^{2+}$ and $Mg^{2+}$ for the EF-3 binding site (*Gutierrez-Ford et al., 2003*). Thus, CHP3 should be present in the $Mg^{2+}$-bound state in a cell at basal $Ca^{2+}$ concentration, ready to respond to $Ca^{2+}$ signals, yet, the exact mechanism is not fully described.

In addition, CHP3 harbors an N-terminal myristoylation site similar to other EFCaBPs such as recoverin and calcineurin B (*Gutierrez-Ford et al., 2003*; *Zaun et al., 2012*). This modification often enhances protein binding to cellular membranes, but it can also stabilize the structure of a protein and/or regulate its function (*Yuan et al., 2020*; *Jiang et al., 2018*). Membrane binding via the myristoyl moiety is usually enforced with either a second lipidation site or clusters of positively charged and/or hydrophobic residues (*Yuan et al., 2020*; *Jiang et al., 2018*). The exposure of the myristoyl moiety from the modified protein and thereby its binding to lipid membranes can be regulated by various signals, for instance by exchange of GDP to GTP (GTP-myristoyl switch in ADP ribosylation factor 1 [ARF1] GTPase) (*Goldberg, 1998*), by phosphorylation (myristoyl/phosphoserine switch in myristoylated alanine-rich C kinase substrate (MARCKS) [*Braun et al., 2000*] and in the C-subunit of protein kinase A [*Gaffarogullari et al., 2011*]) or by pH change (pH-dependent myristoyl-histidine

switch in hisactophilin [*Hanakam et al., 1996*]). In many $Ca^{2+}$ sensor proteins (for instance recoverin, neurocalcin, visinin-like proteins), the myristoyl group becomes accessible to lipid membranes after $Ca^{2+}$ binding (*Lim et al., 2014*). This mechanism is called $Ca^{2+}$-myristoyl switch. However, not all myristoylated EFCaBPs have a $Ca^{2+}$-myristoyl switch. In the neuronal calcium sensor-1 (NCS-1) and in KChIP1, the myristoyl moiety becomes exposed in the presence of lipid membranes even at low $Ca^{2+}$ concentration (*Lim et al., 2014*; *McFerran et al., 1999*; *O'Callaghan and Burgoyne, 2004*; *O'Callaghan et al., 2003*); and it is constantly hidden within the protein core in GCAPs (*Lim et al., 2014*).

Myristoylated CHP1 binds to microsomal membranes in a $Ca^{2+}$-dependent manner indicating the presence of a $Ca^{2+}$-myristoyl switch (*Andrade et al., 2004*), whereas binding of myristoylated CHP3 to lipid membranes has not been reported so far. Co-expression of NHE1 with CHP3 that lacks myristoylation and/or $Ca^{2+}$-binding sites significantly reduced the half-life at the cell surface and the activity of this transporter (*Zaun et al., 2012*). Simultaneous myristoylation and $Ca^{2+}$ binding was suggested to be important for NHE1 stabilization by CHP3 (*Zaun et al., 2012*). Based on these results, the presence of a $Ca^{2+}$-myristoyl switch for CHP3 was proposed (*Gutierrez-Ford et al., 2003*; *Zaun et al., 2012*), though the exposure of the myristoyl group in response to $Ca^{2+}$ binding has not been shown experimentally.

The interaction of CHP3 with NHE1 is an ideal system to analyse the effects of myristoylation and $Ca^{2+}$ binding on CHP3. We previously demonstrated that CHP3 binds at 1:1 ratio to the CHP-binding region of human NHE1 (CBD) with high affinity in the presence of $Mg^{2+}$ (*Fuchs et al., 2018*). At the same time, it was shown by co-immunoprecipitation of CHP3-myc and NHE1-HA that addition of $Ca^{2+}$ increased the amount of the complex formed (*Zaun et al., 2012*). Here, we show that $Ca^{2+}$ and N-terminal myristoylation independently regulate the conformation of CHP3 and its interaction with NHE1 providing the molecular basis for regulation of CHP3 function. This excludes a $Ca^{2+}$-myristoyl switch in CHP3, instead, surface exposure of the myristoyl moiety was triggered by target peptide binding as probed by interaction with liposomes. We named this novel mechanism 'target-myristoyl switch'. Our study provides fundamental mechanistic understanding of the regulation of CHP3 function.

## Results

### Pure fully myristoylated and non-myristoylated untagged CHP3 are functional in binding a single calcium ion

In order to dissect the effects of $Ca^{2+}$ and myristoylation on target and lipid binding of human CHP3, we aimed for pure untagged CHP3 and myristoylated CHP3 (mCHP3). Non-myristoylated CHPs were previously produced with affinity tags and purified by corresponding affinity chromatography (*Gutierrez-Ford et al., 2003*; *Liang et al., 2020*; *Fuchs et al., 2018*). Yet, the peptide tag used for affinity purification of rat CHP1 was shown to interact with the hydrophobic target-binding pocket (*Naoe et al., 2005*) and might interfere with conformational changes. Myristoylated untagged CHP1 had been produced with low yield by co-expression with yeast N-myristoyltransferase (*Timm et al., 1999*). Here, we produced untagged CHP3 and mCHP3, the latter by co-expression with human N-myristoyltransferase 1, and used $Ca^{2+}$-dependent hydrophobic interaction chromatography (HIC) for purification. HIC has been previously used for purification of other EFCaBPs including CaM and calcineurin B (*Tanaka et al., 1984*; *Wei and Lee, 1997*). Combining HIC and gel filtration, we obtained pure untagged proteins, as shown by sodium dodecyl sulfate–polyacrylamide gel electrophoresis (SDS–PAGE) analysis (*Figure 1A*) with an average yield of 9 mg/l of expression culture. Single symmetrical peaks in the elution profiles of analytical gel filtration indicated monodisperse protein preparations of CHP3 and mCHP3. Both proteins form only monomers under reducing conditions in the presence of 2 mM TCEP (*Figure 1—figure supplement 1*). The slightly higher electrophoretic mobility of mCHP3 compared to CHP3 resolved in high-resolution SDS–PAGE analysis already indicated that the protein was covalently modified (*Figure 1A*). To check the degree of myristoylation, we analysed mCHP3 and CHP3 by native mass spectrometry (MS). The measured mass of mCHP3 was increased by ~211 Da (*Figure 1B*, bottom) in comparison to CHP3 (*Figure 1B*, top), which is in agreement with the covalent attachment of one myristoyl group ($M_r$ = 210 Da). Native MS analysis of three independently produced and purified samples confirmed reproducible complete myristoylation of recombinant CHP3. Interestingly, the ion mobility arrival time was shorter for mCHP3 (*Figure 1C*) indicating a more compact shape of mCHP3 compared to CHP3. To reveal the stoichiometry of $Ca^{2+}$

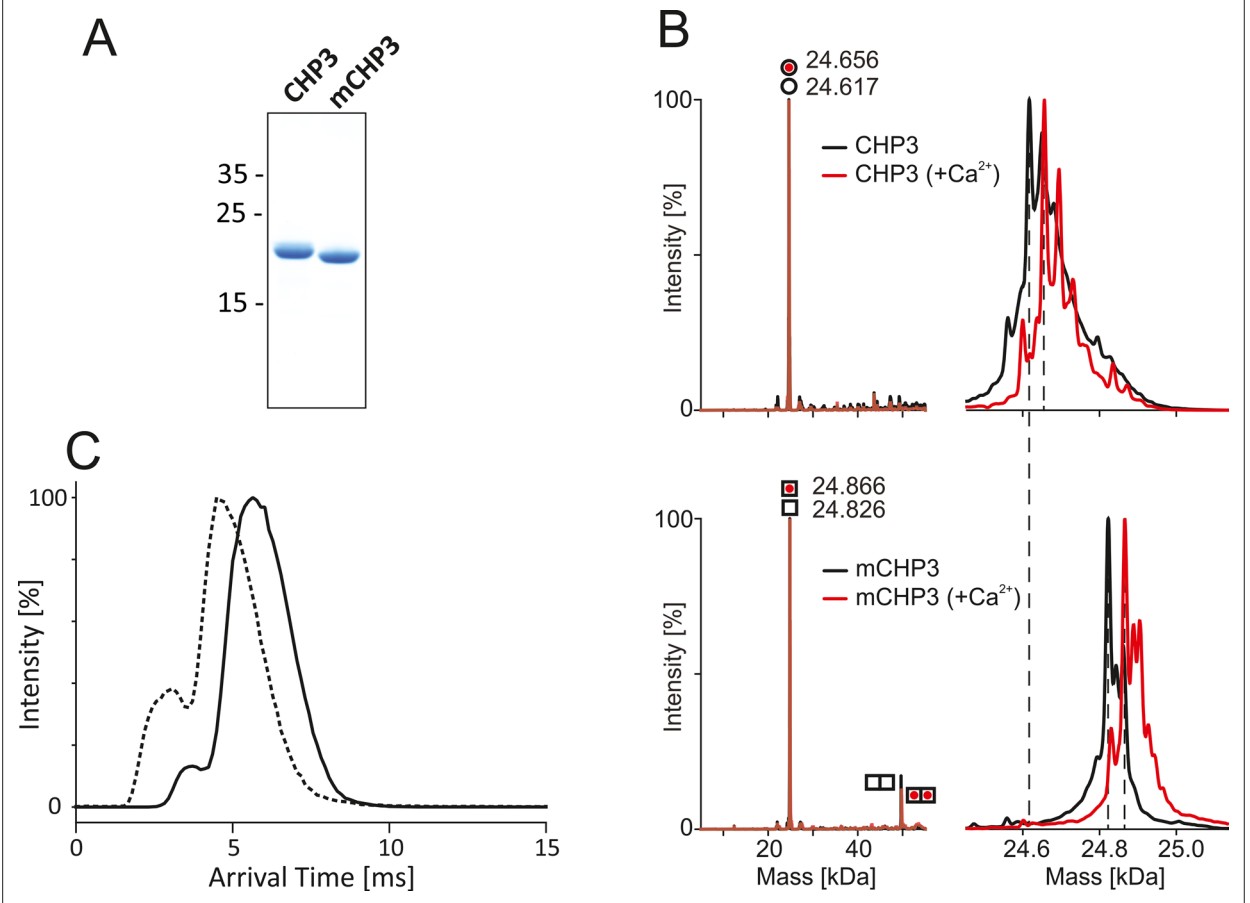

**Figure 1.** Quality control of the purified non-myristoylated CHP3 and myristoylated CHP3 (mCHP3). (**A**) Sodium dodecyl sulfate–polyacrylamide gel electrophoresis (SDS–PAGE) analysis (16%, tricine mini gels) showed a mobility shift of mCHP3 in comparison to CHP3; positions of co-migrated molecular mass standards are indicated in kDa on the left; *Figure 1—source data 1*: Full gel for (A). (**B**) Left, deconvoluted mass spectra of CHP3 (top) and mCHP3 (bottom) show a measured intact protein mass of 24.617 and 24.828 kDa ($\Delta M_r$ = 211 Da), which is in line with the addition of the myristoyl group of 210 Da. For mCHP3, also low abundant dimeric species were observed by native MS analysis. Right, zoom-in spectra in the range of 24.8 ± 0.3 kDa that show differences in masses as indicated by dashed lines. In the presence of $Ca^{2+}$, the measured intact protein mass of CHP3 and mCHP3 was increased by ~40 Da (mass accuracy ± 1 Da), respectively, indicating binding of one $Ca^{2+}$ ion. (**C**) Ion mobility arrival time distributions for the +8 charge state of CHP3 (solid line) and mCHP3 (dashed line). A shoulder with shorter arrival times indicates the presence of a low abundant dimeric species for CHP3 and mCHP3. Arrival times of mCHP3 were reduced by 1.36 ms (±0.37 ms; *n* = 3; p = 0.024) compared to CHP3. See *Figure 1—figure supplement 2* for non-deconvoluted mass spectra showing charge state distributions of CHP3 and mCHP3 measured in the positive ion mode by native MS.

The online version of this article includes the following source data and figure supplement(s) for figure 1:

**Source data 1.** Full gel for *Figure 1A*.

**Figure supplement 1.** Analytical size exclusion chromatography of purified CHP3 (**A**) and mCHP3 (**B**) on Superdex 75 Increase 10/300 GL.

**Figure supplement 2.** Non-deconvoluted mass spectra showing charge state distributions of CHP3 and mCHP3 measured in the positive ion mode by native MS.

binding, we performed native MS analysis of CHP3 and mCHP3 after addition of $Ca^{2+}$. The molecular masses of both proteins were shifted by ~40 Da (*Figure 1B*), which corresponds to the binding of a single $Ca^{2+}$ ion and, thus, documents that both recombinant CHP3 and mCHP3 are functional in respect to $Ca^{2+}$ binding and confirms the presence of one $Ca^{2+}$-binding site (*Gutierrez-Ford et al., 2003*; *Perera et al., 2001*).

## $Ca^{2+}$-induced conformational changes are similar in CHP3 and mCHP3

Next, we analysed the effect of $Ca^{2+}$ binding in the presence of $Mg^{2+}$ and of the N-terminal myristoylation on CHP3 conformation. For this purpose, we optimized the FPH (fluorescence probe

hydrophobicity) assay, previously developed to monitor Ca$^{2+}$-induced conformational changes of CHPs (*Liang et al., 2020*), by using the dye ProteOrange (Lumiprobe) at defined micromolar concentration (see Materials and methods). In this assay, the fluorescence of the dye strongly increases upon its binding to hydrophobic protein surfaces (*Niesen et al., 2007*).

In the Mg$^{2+}$-bound state, CHP3 and mCHP3 showed low fluorescence (*Figure 2—figure supplement 1A*). After Ca$^{2+}$ addition, the fluorescence intensity increased by 30% and 40% for CHP3 and mCHP3, respectively, reflecting an increase in hydrophobicity (*Figure 2A*). Ca$^{2+}$ removal by EGTA reduced the fluorescence to the initial level, indicating the reversibility of Ca$^{2+}$ binding and of the respective conformational changes. Depletion of both Mg$^{2+}$ and Ca$^{2+}$ by EDTA turned the protein into the non-physiological apo-state, the fluorescence was in between that of the Mg$^{2+}$- and Ca$^{2+}$-bound states. The proteins appeared to be destabilized in the apo-state and could not be reverted into the functional form by Ca$^{2+}$ addition (*Figure 2A*). Interestingly, the N-terminal myristoylation did not affect the conformational changes of CHP3. As a control, we probed the hydrophobicity of recoverin, the prototypic protein with a classical Ca$^{2+}$-myristoyl switch (*Ames et al., 1997*; *Figure 2—figure supplement 1B*). Ca$^{2+}$ induced similar changes of non-myristoylated recoverin as of CHP3 in the FPH assay, whereas hydrophobicity of the myristoylated protein increased much stronger (about 3.5-fold) in response to Ca$^{2+}$ binding, which most likely resulted from the exposure of the myristic group.

To prove that changes in the dye-mediated fluorescence observed in the FPH assay were indeed caused by conformational changes, we probed the intrinsic tryptophan fluorescence of CHP3 and mCHP3. According to the 3D model of CHP3 predicted with AlphaFold2.0 (*Varadi et al., 2022*), the single tryptophan residue (Trp191) is located in CHP3's hydrophobic target-binding pocket at the protein surface (*Figure 2C*) in line with the measured emission maxima of CHP3 and mCHP3 of ~330 nm (*Figure 2—figure supplement 2*). Intensity and emission maximum of intrinsic tryptophan fluorescence depend on the local environment of the residue (*Vivian and Callis, 2001*; *Eftink, 2000*). Ca$^{2+}$ addition decreased the intrinsic fluorescence of CHP3 (*Figure 2B*) reflecting conformational changes accompanied with changes in the local environment of Trp191 in the hydrophobic pocket. This is in line with an increase of CHP3 hydrophobicity observed with the FPH assay. These conformational changes monitored by intrinsic fluorescence were also reverted by Ca$^{2+}$ removal (EGTA addition) (*Figure 2B*). Only for mCHP3, the removal of both Mg$^{2+}$ and Ca$^{2+}$ caused irreversible changes in the intrinsic fluorescence.

In order to evaluate whether myristoylation affects the Ca$^{2+}$-binding affinity of CHP3, we determined the EC50 values for binding of Ca$^{2+}$ to CHP3 and mCHP3 in the presence of Mg$^{2+}$ using the FPH assay. Addition of Ca$^{2+}$ at saturating concentration increased the fluorescence (*Figure 2A*). We now titrated Ca$^{2+}$ concentration from 0.3 μM to 8 mM and measured fluorescence using biological and technical replicates (*Figure 2D, E*). We determined Ca$^{2+}$ EC50 values performing a global fit for all data. The difference between EC50's obtained for CHP3 (161.1 [112.6; 233.8] μM; *Figure 2D*) and mCHP3 (152.6 [113.7; 209.7] μM; *Figure 2E*) is insignificant, indicating that myristoylation does not affect the Ca$^{2+}$-binding properties of CHP3.

Combining the data of the FPH assay and of the intrinsic tryptophan fluorescence, we conclude that CHP3 undergoes reversible Ca$^{2+}$-induced conformational changes as typical for a Ca$^{2+}$ sensor protein. The hydrophobic target-binding pocket of CHP3 is occluded in the Mg$^{2+}$-bound state (closed conformation), and becomes exposed to the environment upon Ca$^{2+}$ binding (open conformation).

## Ca$^{2+}$ binding and myristoylation independently affect thermal stabilities of CHP3 and its complex with NHE1 target peptide

To further dissect the effects of Ca$^{2+}$ and N-terminal myristoylation on CHP3, we probed the thermal stability of CHP3 alone and in complex with the CHP-binding domain of NHE1 (CBD, NHE1 residues 525–545) using nano-differential scanning fluorimetry (nanoDSF). To obtain the CHP3:CBD complex used for nanoDSF, we co-expressed CHP3 and CBDHis using the pETDuet-1 system and purified the resulting complex. For the myristoylated complex (mCHP3:CBD), simultaneous co-expression of three proteins (CHP3, CBDHis, and human *N*-myristoyltransferase 1 (NMT)) from a modified pETDuet-1 vector was performed, and the resulting complex was purified. Myristoylation of CHP3 in the complex was confirmed by electrospray ionization (ESI)-TOF mass spectrometric analysis. Free CHP3 had the highest thermal stability in the Mg$^{2+}$-bound state ($T_m^{app}$ 70.1 ± 0.3°C), and Ca$^{2+}$ binding reduced the melting point to 66.9 ± 0.2°C (CHP3 in *Figure 3*). In the presence of both ions, the Ca$^{2+}$ effect

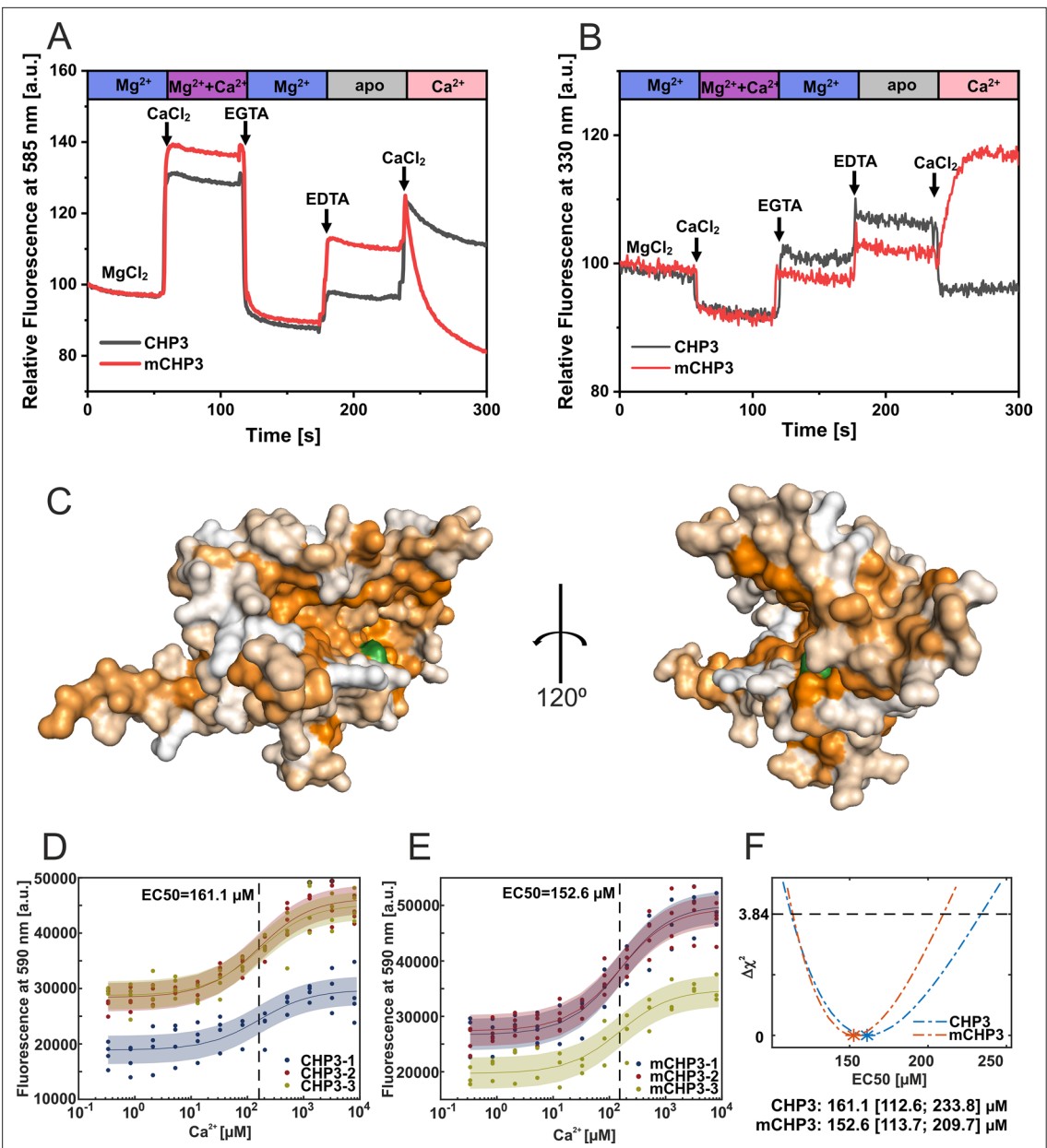

**Figure 2.** Ca²⁺-induced conformational changes in CHP3 and mCHP3. (**A**) Kinetic fluorescence probe hydrophobicity (FPH) assay. Fluorescence of dye bound at hydrophobic protein surfaces was monitored at $\lambda_{em}$ = 585 nm (excitation $\lambda_{ex}$ = 470 nm) and at 22°C. Protein (1.5 µM) was prepared in the Mg²⁺-bound state (2 mM MgCl₂, 1 mM EGTA). CaCl₂, EDTA, and EGTA were sequentially added as indicated. First, 2 mM CaCl₂ was added and then chelated with addition of 3 mM EGTA. Next, 3 mM EDTA was added to remove both divalent ions (CHP3 in apo-state) followed by another addition of 4 mM CaCl₂. (**B**) Intrinsic tryptophan fluorescence was monitored at $\lambda_{em}$ = 330 nm (excitation $\lambda_{ex}$ = 280 nm) at 22°C. Protein (2.5 µM) in the Mg²⁺-bound state was used and the additions were performed as described above for the FPH assay. (**C**) AlphaFold2.0 model (**Varadi et al., 2022**) of CHP3 in surface presentation, with surface coloured for hydrophobicity (**Eisenberg et al., 1984**). The model most likely resembles the open or target-bound conformation. The single tryptophan residue highlighted in green (Trp191) is located in the hydrophobic target-binding pocket. EC50 values for binding of Ca²⁺ to CHP3 (**D**) and mCHP3 (**E**) determined with FPH assay. Fluorescence of samples with CHP3 or mCHP3 at different Ca²⁺ concentrations was measured at 590 nm in the presence of 2 mM MgCl₂. Three biological replicates (shown in different colours) with three to four technical replicates each for CHP3 and mCHP3 were measured. The data were fitted with Hill equation using global non-linear regression. (**F**) 95% confidence intervals (Δχ² of 3.84) of EC50 values for binding of Ca²⁺ calculated with profile likelihood method. EC50 values (asterisks) are shown with confidence intervals in square brackets below the graph.

The online version of this article includes the following figure supplement(s) for figure 2:

**Figure supplement 1.** Conformational changes of CHP3, mCHP3, recoverin, and myristoylated recoverin monitored with the fluorescence probe hydrophobicity (FPH) assay.

*Figure 2 continued on next page*

*Figure 2 continued*

**Figure supplement 2.** Intrinsic tryptophan fluorescence spectra of CHP3 (**A**) and mCHP3 (**B**).

appeared to be dominant with a melting temperature ($T_m^{app}$) of 65.0 ± 0.9°C. In line with the results of the FPH assay, which indicated a destabilized apo-state, the latter showed strongly reduced thermal stability ($T_m^{app}$ 56.1 ± 0.5°C). Notably, binding of the target peptide CBD strongly increased the thermal stability for the $Ca^{2+}$-bound and apo-states (compare CHP3 and CHP3:CBD in *Figure 3*), with about 12°C in the presence of $Ca^{2+}$, 13°C in the presence of $Ca^{2+}$ plus $Mg^{2+}$ and about 11°C for the apo-state. Thus, the CHP3:CBD complex showed the highest thermal stability when $Ca^{2+}$ was bound. Surprisingly, CBD binding did not have any effect on the thermal stability of the $Mg^{2+}$-bound state.

The N-terminal myristoylation decreased the $T_m^{app}$ of CHP3 in all states to the same degree (9.8°C for $Mg^{2+}$-bound and 10.6°C for $Ca^{2+}$-bound states, 10.9°C in the presence of both ions, 4.5°C for the already destabilized apo-state) (compare CHP3 and mCHP3 in *Figure 3*). A similar destabilizing effect of the myristoylation was observed for the mCHP3:CBD complex with a decrease of $T_m^{app}$ in the range from 6.4°C for the $Ca^{2+}$-bound state to 3.2°C for the apo-state (compare CHP3:CBD and mCHP3:CBD in *Figure 3*). Thus, the N-terminal myristoylation lowers the stability of CHP3 independently of the bound ion and of the target binding that means independent of the conformation.

## $Ca^{2+}$ binding to CHP3 allows more effective cleavage within EF-2, whereas CBD binding drastically increases proteolytic stability

In order to pinpoint flexible regions of CHP3 in a given conformation and to probe whether these regions are affected by myristoylation, ion and target binding, we performed limited trypsinolysis of CHP3, mCHP3, and their complexes with CBD in the presence of $Ca^{2+}$, $Mg^{2+}$, or EDTA.

CHP3 was readily proteolysed by trypsin with nearly one half of the protein already degraded after 5 min and almost no full-length protein remained after 60-min incubation (*Figure 4A*, top, FL). Trypsin cleavage sites (Arg and Lys residues) are distributed all over the CHP3 amino acid sequence (*Figure 4—figure supplement 1A*), yet productive cleavage requires not only availability of the site but also flexibility of the peptide chain at the cleavage site to adapt to the active site of the protease (*Teilum et al.,*

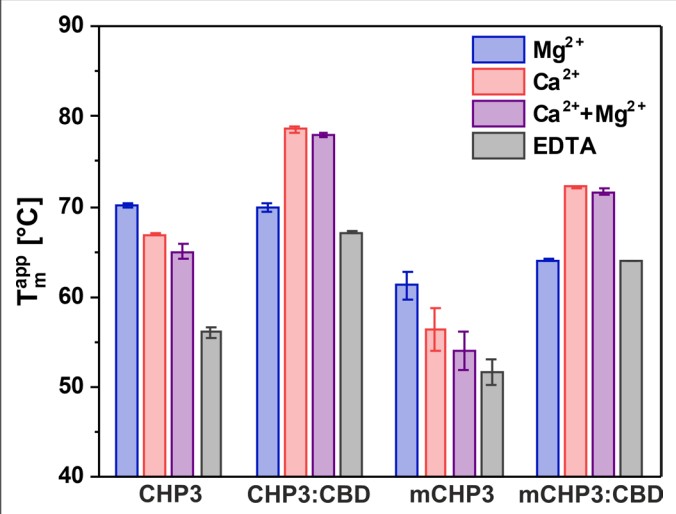

**Figure 3.** $Ca^{2+}$ and target peptide binding affect the thermal stability of CHP3 and mCHP3. Thermal stabilities of free proteins and complexes with the target peptide CBD were measured with nanoDSF in the presence of either 10 mM $Mg^{2+}$, $Ca^{2+}$, $Ca^{2+}$ + $Mg^{2+}$, or EDTA. Temperatures of thermal unfolding (apparent $T_m$) are shown as mean ± standard deviation (SD) of three independent biological replicates. Raw nanoDSF traces are shown in *Figure 3— figure supplement 1*.

The online version of this article includes the following figure supplement(s) for figure 3:

**Figure supplement 1.** Thermal unfolding of CHP3 (**A**), mCHP3 (**B**), CHP3:CBD (**C**), and mCHP3:CBD (**D**) shown as raw nanoDSF traces (left) and first derivatives (right) in the presence of either $Mg^{2+}$, $Ca^{2+}$, $Mg^{2+}$ + $Ca^{2+}$, or in the absence of both ions (EDTA).

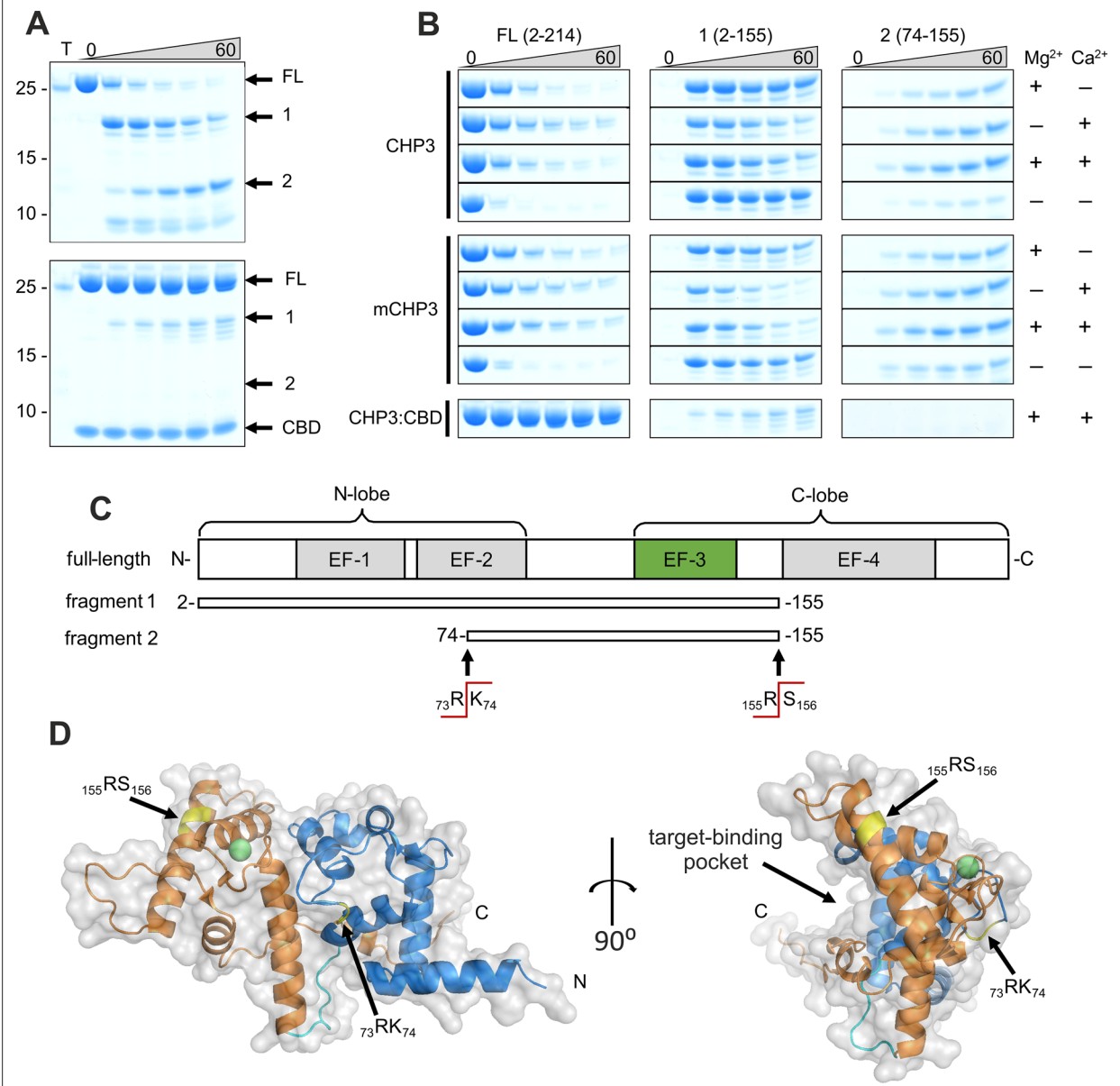

**Figure 4.** Ca²⁺-binding and complex formation change the accessibility of trypsin cleavage sites in CHP3 and mCHP3. (**A**) Time-dependent (0–60 min) limited proteolysis (trypsin) of CHP3 (top) and the complex of CHP3:CBD (bottom) in the presence of both Mg²⁺ and Ca²⁺. Positions of full-length protein (FL) and two major proteolytic fragments (1 and 2) as well as CBD are indicated on the right of the Coomassie-stained SDS–PAGE gel, positions of co-separated molecular mass standards (mass in kDa) – on the left; the sample containing only trypsin was loaded on the first lane (T); **Figure 4— source data 1**: Full gels of (A). (**B**) Time-dependent limited proteolysis of CHP3 and mCHP3 in the presence of Mg²⁺, Ca²⁺, both ions or in the absence of them. Sections of the gel with bands corresponding to the full-length protein (FL) and two major proteolytic fragments (1 and 2) are shown. Nearly no degradation was observed for CHP3 and mCHP3 in the complex with CBD in all conditions (Mg²⁺ + Ca²⁺ condition is presented here, other gels are shown in **Figure 4—figure supplement 1B** and its source data). (**C**) Schematic representation of full-length CHP3 with indication of N- and C-lobes, four EF-hand motifs (active EF-3 is highlighted in green). Proteolytic fragments 1 and 2 and trypsin cleavage sites were identified by mass spectrometry. (**D**) Combined ribbon and surface presentation of the CHP3 AlphaFold2.0 model (**Varadi et al., 2022**) with N- and C-lobes shown in blue and orange, respectively, and the connecting CHP-loop in cyan; the two major trypsin cleavage sites are highlighted in yellow and Ca²⁺ ion as a green sphere. The Ca²⁺ position in EF-3 was modelled by superimposition of the CHP3 model with the CHP1 X-ray structure, pdb ID 2ct9 (**Andrade et al., 2004**).

The online version of this article includes the following source data and figure supplement(s) for figure 4:

**Source data 1.** Full gels for **Figure 4A**.

**Figure supplement 1.** Limited trypsinolysis of CHP3 and mCHP3.

**Figure supplement 1—source data 1.** Full gels for **Figure 4—figure supplement 1B**.

**Figure supplement 2.** Multiple sequence alignment of human CHP1, CHP2, and CHP3.

*2009*). During the reaction, we observed the appearance of two major fragments (labelled 1 and 2 in *Figure 4A*). We analysed those fragments with MS to determine exact masses and precisely locate the cleavage sites (*Figure 4—figure supplement 1C*). Fragment 1 has a mass of 17.92 kDa corresponding to residues 2–155 of CHP3 (UniProtID Q96BS2-1) lacking the C-terminal part (*Figure 4C*). Fragment 2 has a mass of 9.63 kDa and includes residues 74–155. Thus, it derived from a cleavage of fragment 1 within the predicted EF-2 (*Figure 4C*). Noteworthy, both fragments still contain the single functional EF-hand of CHP3 (EF-3). Target peptide binding nearly completely prevented the trypsin cleavage, and the intensity of the full-length CHP3 band was only slightly reduced even after 60-min incubation (*Figure 4A*, bottom). Next, we compared the proteolytic stability of CHP3 in different states. In the presence of only $Ca^{2+}$, $Mg^{2+}$, or both ions together, the degradation rate of the full-length protein (FL) was comparable, whereas the apo-state degraded much faster (*Figure 4B*, FL). Fragment 1 appeared already after 5 min and degraded further over time (*Figure 4B*, 1), its cleavage was more pronounced in the presence of $Ca^{2+}$ and was accompanied with an increase of fragment 2 (*Figure 4B*, 2). This indicates that the cleavage site located in EF-2 becomes more available for the protease in the open conformation ($Ca^{2+}$ bound). N-terminal myristoylation did not affect the proteolytic stability of the full-length protein, whereas it slightly reduced the stability of fragment 1 in all states. In addition, a second minor band below fragment 1 appeared for mCHP3 (*Figure 4B*). The small differences in the tryptic cleavage of CHP3 and mCHP3 indicate a local destabilizing effect of the myristoyl moiety, in line with the lower thermal stability of mCHP3 compared to CHP3. The complex formation with CBD drastically increased the proteolytic stability of both CHP3 and mCHP3, as nearly no degradation occurred under all analysed experimental conditions (*Figure 4B*, CHP3:CBD and *Figure 4—figure supplement 1B*).

We mapped both major cleavage sites on the 3D model of CHP3 predicted with AlphaFold2.0 (*Varadi et al., 2022*; *Figure 4D*). This model closely resembles the structures of CHP1 and CHP2 in complex with CBD (*Ammar et al., 2006*; *Mishima et al., 2007*), of CHP1 in complex with full-length NHE1 (*Dong et al., 2021*) and of CHP1 with an artificial C-terminal helix bound in the target-binding

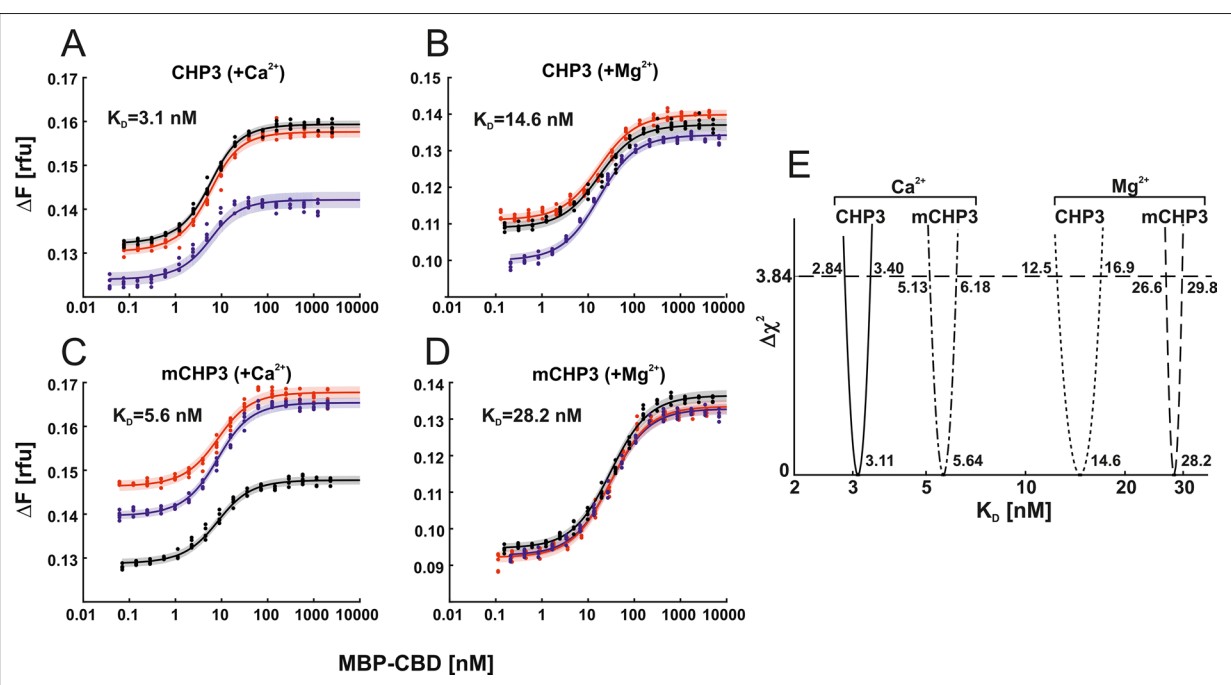

**Figure 5.** $Ca^{2+}$ binding and myristoylation independently affect the interaction between CHP3 and MBP-CBD as measured with microscale thermophoresis (MST). The interaction of CHP3 (**A, B**) and mCHP3 (**C, D**) with MBP-CBD was measured with three biological replicates shown in different colours (each with five technical replicates) in the presence of either $Ca^{2+}$ (**A, C**) or $Mg^{2+}$ (**B, D**). The combined data from individual experiments were fitted with a one-site binding model using global non-linear regression. (**E**) 95% confidence intervals ($\Delta\chi^2$ of 3.84) of $K_D$'s for CHP3:CBD and mCHP3:CBD calculated with profile likelihood method. Raw MST traces are shown in *Figure 5—figure supplement 1*.

The online version of this article includes the following figure supplement(s) for figure 5:

**Figure supplement 1.** Normalized microscale thermophoresis (MST) fluorescence timetraces of CHP3 and mCHP3 with MBP-CBD in the presence of either $Mg^{2+}$ or $Ca^{2+}$.

pocket (*Kennedy et al., 1996*), that is structures of target-bound CHPs. All these structures represent the $Ca^{2+}$-bound state. The cleavage site R155/S156 is located in this model at the N-side of the incoming α-helix of EF-4 and though surface exposed the α-helical location may hamper the proteolytic digest (*Fontana et al., 1997*). The other site, R73/K74 is located in the middle of the EF-2 loop (non-functional $Ca^{2+}$-binding loop) (*Figure 4D*). Though the site appears to be surface exposed in the model, one should note that CHP3 has a 9-amino acid long insertion in that position as compared to CHP1 and CHP2 (*Figure 4—figure supplement 2*). The structure of CHP3 may deviate from the model and information for other conformations is lacking. Clearly, limited proteolysis showed that both sites are accessible and sufficiently flexible for productive cleavage in the free CHP3, yet become protected against cleavage upon complex formation. Both major cleavage sites are highly unlikely to be covered by CBD in the complex, as they are facing away from the target-binding pocket (*Figure 4D*). Thus, the drastic changes in the proteolytic stability indicate reduced flexibility in EF-2 and EF-4, which could be caused by structural rearrangements of CHP3 upon target peptide binding.

## $Ca^{2+}$ binding and the N-terminal myristoylation independently modulate the affinity of CHP3 for NHE1

To analyse how the conformational changes described above affect CHP3 function, we measured affinities of CHP3 and mCHP3 for the target peptide CBD with microscale thermophoresis (MST) in $Ca^{2+}$- and in $Mg^{2+}$-bound states. We used the system established earlier for quantification of binding affinities of CHP isoforms (CHP1, CHP2, and CHP3) to NHE1, in which the target peptide CBD was fused to maltose-binding protein (MBP-CBD) for its stabilization (*Liang et al., 2020*; *Fuchs et al., 2018*). We conducted the MST experiments with three biological replicates per sample and five individual titration series (technical replicates) per biological replicate (*Figure 5*). The dissociation constant ($K_D$) for a given sample was calculated with a global parameter estimation procedure including all biological and technical replicates. We fitted the mass action law to the MST data points using non-linear regression. Confidence intervals were assessed for each $K_D$ value by calculating profile likelihoods (error surface plots) as described elsewhere (*Kreutz et al., 2013*; *Scheuermann et al., 2016*; *Figure 5E*). Comparison of the resulting $K_D$ values showed that the effects of $Ca^{2+}$ binding and myristoylation on CHP3:CBD interaction are independent. Namely, $Ca^{2+}$ binding increased affinities for CBD of both non-myristoylated (CHP3:CBD, $K_D$ = 3.1 [2.8; 3.4] nM) and myristoylated CHP3 (mCHP3:CBD, $K_D$ = 5.6 [5.1; 6.2] nM) by about fivefold as compared to the $Mg^{2+}$-bound state (CHP3:CBD, $K_D$ = 14.6 [12.5; 16.9] nM; mCHP3:CBD, $K_D$ = 28.2 [26.6; 29.8] nM) (*Figure 5A–D*). In turn, for both $Ca^{2+}$- and $Mg^{2+}$-bound states, myristoylation decreased the affinity twofold (*Figure 5*). Hence, CHP3 binds to NHE1 with nanomolar affinity in both open and closed conformation and independently of its modification, however, $Ca^{2+}$ binding and myristoylation modulates the affinity.

## Target peptide and $Ca^{2+}$ binding regulate the interaction of CHP3 with lipid membranes

As we showed that the effects of $Ca^{2+}$ and myristoylation on conformational changes of CHP3 and mCHP3 are independent, the hypothesis of a $Ca^{2+}$-myristoyl switch for CHP3 (*Gutierrez-Ford et al., 2003*; *Zaun et al., 2012*) had to be questioned. Therefore, we analysed binding of CHP3 and mCHP3 to POPC:POPS (3:1 molar ratio) liposomes with a co-sedimentation assay in the presence of either $Mg^{2+}$ or $Ca^{2+}$, and compared to the binding of recoverin, used as control protein for a $Ca^{2+}$-myristoyl switch. In line with previous studies (*Zozulya and Stryer, 1992*), non-myristoylated recoverin (Rec) only weakly interacted with liposomes in both $Mg^{2+}$- and $Ca^{2+}$-bound states, whereas the interaction of myristoylated protein (mRec) increased about fourfold upon $Ca^{2+}$ addition (*Figure 6A, B*). In the $Mg^{2+}$-bound state, CHP3 interacted weakly with liposomes similar to recoverin, only a small protein amount co-sedimented with liposomes. In contrast to recoverin, $Ca^{2+}$ addition increased the binding of non-myristoylated CHP3 (about 2.5 times) to liposomes, whereas myristoylation did not further increase this binding (*Figure 6A, B*). Interestingly, the binding of the target peptide CBD substantially changed the interaction of CHP3 with liposomes. In both the absence and the presence of $Ca^{2+}$, only a small amount of the CHP3:CBD complex was associated with liposomes. However, CHP3 myristoylation increased the binding in both conditions about fourfold (*Figure 6A, B*). In the presence of $Ca^{2+}$, the binding of mCHP3:CBD was even slightly higher than without it, however the difference was not significant. These results indicate the following mode of interaction. The myristic moiety is located

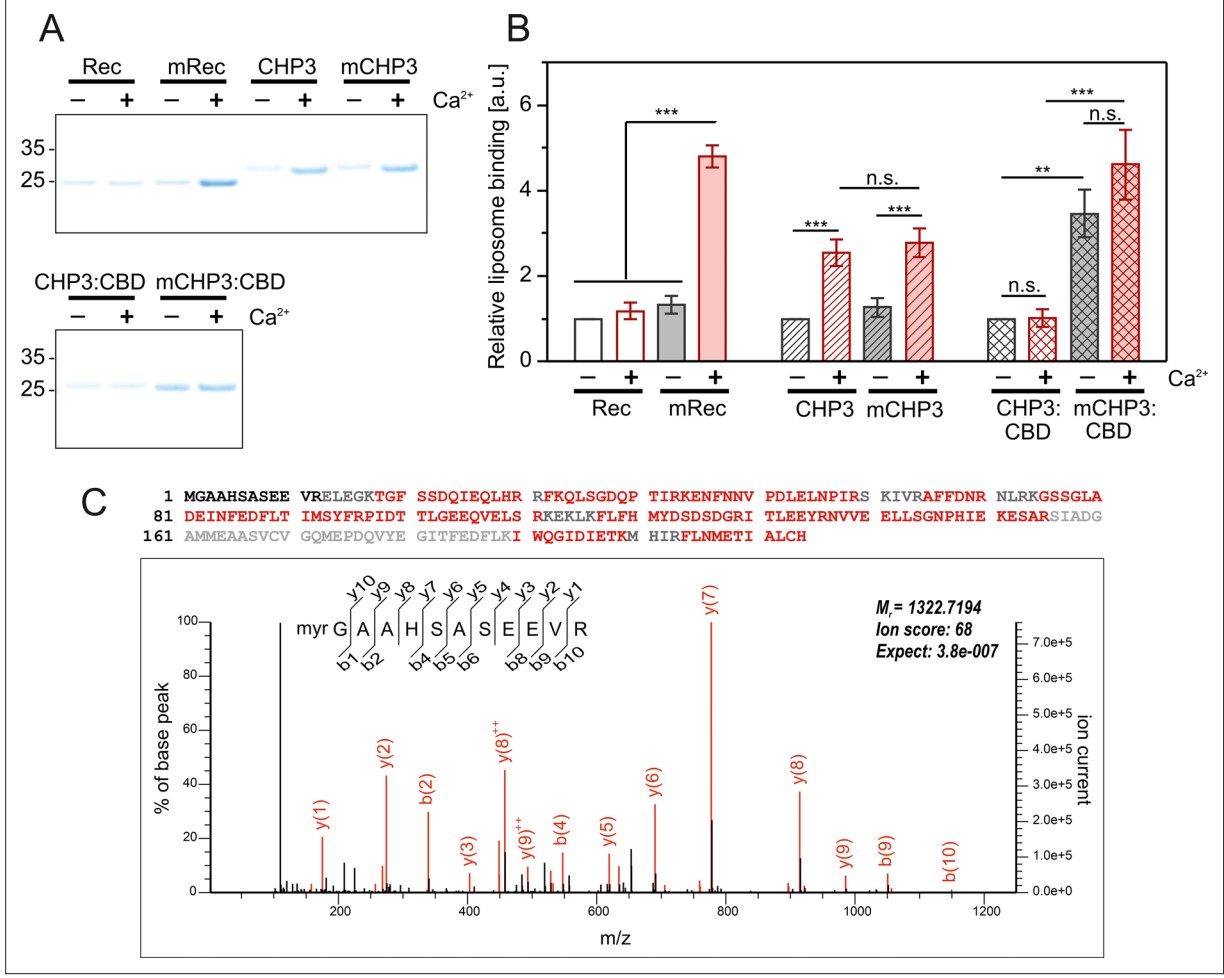

**Figure 6.** The interaction of CHP3 and mCHP3 with liposomes is regulated by $Ca^{2+}$ and target peptide binding. Proteins were co-sedimented with POPC:POPS (3:1 molar ratio) liposomes in the presence of either 2 mM $Mg^{2+}$ or 2 mM $Mg^{2+}$ + 2 mM $Ca^{2+}$ at 24°C. Non-myristoylated and myristoylated recoverin (Rec and mRec, respectively) were used as $Ca^{2+}$-myristoyl switch control proteins. (**A**) Amount of proteins co-sedimented with liposomes was analysed with SDS–PAGE (4–12%, Bis-Tris). (**B**) Quantification of protein-liposome binding based on the densitometry of bands (SDS–PAGE shown in (**A**)) corresponded to the co-sedimented proteins with three biological (CHP3 and complexes) or technical (recoverin) replicates. Values were normalized to the respective non-myristoylated protein in the $Mg^{2+}$-bound state. Data are shown as mean ± standard deviation (SD), one-way analysis of variance (ANOVA) with Tukey post-test was performed for mean comparison (statistical significance: n.s. – p > 0.05, **p < 0.01, ***p < 0.001). (**C**) N-terminal myristoylation of target-associated CHP3 in mouse brain. LC–MS/MS analysis of a trypsin-digested size fraction of solubilized mouse brain membrane containing NHE1-associated CHP3 (see Materials and methods). Upper panel: high coverage of the mouse CHP3 primary sequence by MS/MS-identified peptides (in red; black: sequences not identified, grey: sequences not accessible to our MS analysis) without inclusion of N-myristoyl modification. Lower panel: MS/MS spectrum from the same measurement assigned to the myristoylated tryptic N-terminal peptide of CHP3. Note that no other forms of the N-terminal peptide were detectable in error-tolerant search. *Figure 6—source data 1*: Full gels for (A) and replicates; raw data of band densitometry and ANOVA test p-values.

The online version of this article includes the following source data and figure supplement(s) for figure 6:

**Source data 1.** Full gels for *Figure 6A* and replicates; raw data of band densitometry and analysis of variance (ANOVA) test p-values for *Figure 6B*.

**Figure supplement 1.** Sequence coverage and N-terminal myristoylation of endogenous CHP3.

in the protein core of target-free CHP3 and is not accessible to lipid membranes in the presence of $Ca^{2+}$. Most likely, other protein regions are involved in membrane binding in this case. Target peptide binding causes conformational rearrangements with displacement of the myristic moiety from the protein core to the environment, which leads in consequence to an increased association of CHP3 and its target to lipid membranes.

Finally, we investigated the N-terminal myristoylation status of membrane associated CHP3 in vivo using liquid chromatography coupled mass spectrometry (LC–MS/MS). In three different preparations,

namely (1) size fractionated complexes from mouse brain membranes (*Figure 6C*), (2) integral membrane proteins isolated from mouse platelets (i.e. after carbonate extraction, *Figure 6—figure supplement 1*), and (3) anti-NHE1-affinity purification from solubilized mouse brain membranes (data not shown) we identified the vast majority of MS-accessible CHP3 peptides, but only the myristoylated form of the N-terminal peptide. Together, this suggests that myristoylated CHP3 is both NHE1 associated and membrane anchored in agreement with a target-induced exposure and membrane integration of the N-terminal myristoyl moiety.

## Discussion

$Ca^{2+}$-induced conformational changes are a hallmark of $Ca^{2+}$-sensing EFCaBPs, which control the interaction with and the function of specific target proteins (*Burgoyne, 2007*). Our in vitro experiments show that $Ca^{2+}$ binding induces conformational changes in CHP3, thus it fulfills this prerequisite to respond to $Ca^{2+}$ signals in the cell. Based on the reported affinities of CHP3 for $Ca^{2+}$ and $Mg^{2+}$ (*Gutierrez-Ford et al., 2003*), CHP3 should bind $Mg^{2+}$ at resting $Ca^{2+}$ concentration of about 0.1 µM in the cell (*Roderick and Cook, 2008*; *Bootman and Bultynck, 2020*) and it could bind $Ca^{2+}$ when intracellular $Ca^{2+}$ concentration rises up to 1 µM during signalling events (*Bootman and Bultynck, 2020*). As CHP3 can be N-terminally myristoylated, it was proposed to be regulated with a so-called $Ca^{2+}$-myristoyl switch mechanism (*Gutierrez-Ford et al., 2003*; *Zaun et al., 2012*), in which $Ca^{2+}$ binding induces the exposure of the myristic moiety to the environment. In contrast to this hypothesis, we described in the present study independent effects of $Ca^{2+}$ binding and N-terminal myristoylation on CHP3 conformation and on the interaction with its target protein NHE1. We show that target peptide and not $Ca^{2+}$ binding causes strong association of the myristoylated CHP3 to lipid membranes.

First, we optimized the protein production and purification to obtain pure myristoylated and non-myristoylated CHP3 devoid of affinity tags, a prerequisite for further quantitative biochemical and biophysical analysis. With the expression system based on that described for the production of N-myristoylated Nef protein of HIV-1 (*Glück et al., 2010*), we obtained preparative amounts of fully myristoylated CHP3. Stoichiometric myristoylation, which is still challenging for recombinant EFCaBPs (*Vladimirov et al., 2020*), was shown with native MS. The latter was also used to demonstrate that both CHP3 and mCHP3 bind a single $Ca^{2+}$, and are thus functional EFCaBPs, in agreement with the results of equilibrium dialysis shown earlier (*Gutierrez-Ford et al., 2003*). Using the highly pure proteins, we showed with FPH and intrinsic fluorescence assays reversible conformational changes in CHP3 between a more hydrophobic $Ca^{2+}$-bound state typical for an open conformation and a less hydrophobic $Mg^{2+}$-bound state reflecting a closed conformation (*Figure 7*). Such conformational

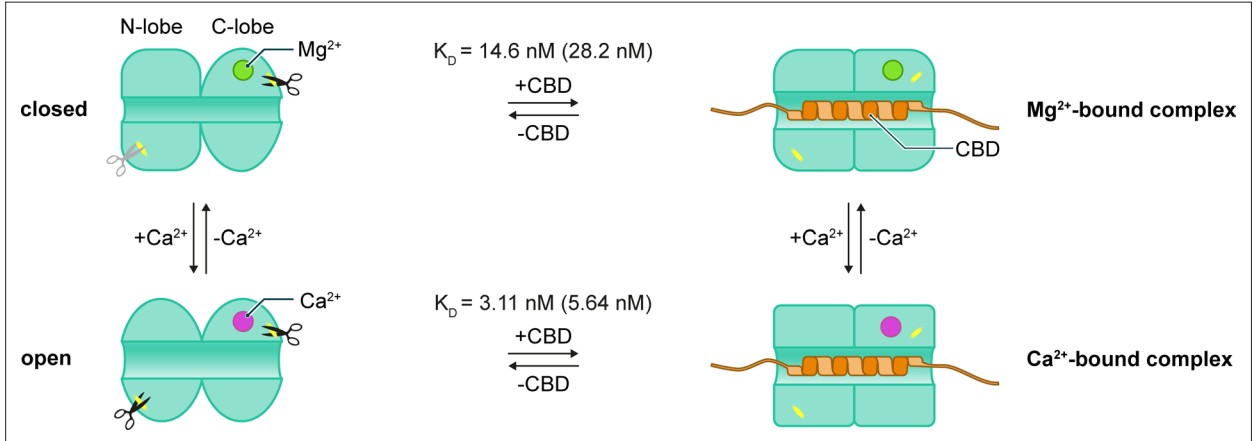

**Figure 7.** Conformation of CHP3 is controlled by $Ca^{2+}$ and target peptide (NHE1 CBD) binding. $Ca^{2+}$ (magenta) replaces $Mg^{2+}$ (green) in EF-3 and induces the transition from the closed to the open conformation. $Ca^{2+}$ binding affects the local flexibility in the N-lobe accelerating the tryptic cleavage and reduces the thermal stability of CHP3. In both, $Ca^{2+}$- and $Mg^{2+}$-bound states, CHP3 binds the target peptide CBD with nanomolar affinity ($K_D$ values indicated, in brackets for myristoylated CHP3). Complex formation has no effect on the thermal stability of the $Mg^{2+}$-bound state (closed conformation), whereas it strongly enhances thermal stability of the $Ca^{2+}$-bound state (open conformation). Independent of the bound ion, both trypsin cleavage sites are protected from proteolysis in the CHP3:CBD complex.

changes are typical for $Ca^{2+}$ sensor EFCaBPs including calmodulin, recoverin, neuronal calcium sensor protein (NCS1), and others, and serve as a conformational switch for binding and/or regulation of a target protein (*Chazin, 2011*). We characterized different CHP3 conformations in respect to thermal and proteolytic stability. We revealed a stabilizing effect of $Mg^{2+}$ on target-free CHP3 and mCHP3 with increased thermal stability. As we observed that CHP3 is not stable in the apo-state, we conclude that $Mg^{2+}$ binding is important to stabilize the closed conformation of target-free protein in the cell at resting $Ca^{2+}$ level. Stabilization by $Mg^{2+}$ binding has been shown for other EFCaBPs including calmodulin and troponin C (*Gifford and Vogel, 2013*; *Grabarek, 2011*). Further, replacement of $Mg^{2+}$ with $Ca^{2+}$ at EF-3 (C-lobe) lowered the thermal stability of CHP3 and mCHP3 and enhanced proteolytic cleavage in EF-2 (N-lobe) suggesting an increase in local flexibility. Thus, $Ca^{2+}$-induced conformational changes involve not only the C-lobe of CHP3 but also its N-lobe.

The $Ca^{2+}$-induced open conformation with lower thermal stability and increased local flexibility appears to provide the molecular basis for CHP3 to interact more readily with its target(s) (*Figure 7*). Our data support this hypothesis. Firstly, the binding of the target peptide resulted in a drastic increase of CHP3 thermal and proteolytic stabilities. However, the thermal stability increased only in the presence of $Ca^{2+}$, whereas for the $Mg^{2+}$-bound state it remained unchanged. These characteristics of the $Ca^{2+}$-induced open conformation may lower the energy barrier for binding of the target peptide. Secondly, in line with co-immunoprecipitation of tagged NHE1 and tagged CHP3 from transfected AP-1 cells (*Zaun et al., 2012*), we detected binding between CHP3 and CBD in both, $Ca^{2+}$- and $Mg^{2+}$-bound states (see *Figure 5*). The quantitative analysis with MST showed that the affinity of the $Ca^{2+}$-bound state was about five times higher, clearly indicating that $Ca^{2+}$ controls CHP3 function.

Though all CHP isoforms bind NHE1 with nanomolar affinities in the absence and presence of $Ca^{2+}$ (*Liang et al., 2020*; *Fuchs et al., 2018*), intracellular $Ca^{2+}$ signals might regulate NHE1 function (e.g. activity or stability) via CHPs affecting the conformation of the complex. Regulation of the activity by $Ca^{2+}$ binding to the regulatory subunit was described for calcineurin, a phosphatase consisting of the catalytic and regulatory subunits calcineurin A (CnA) and B (CnB), respectively. Similar to the CHP3:NHE1 interaction, CnB forms a stable complex with CnA independently of $Ca^{2+}$ binding. Nevertheless, $Ca^{2+}$ binding induces conformational changes of CnB resulting in the release of the calmodulin-binding domain of CnA and allowing its activation by calmodulin (*Creamer, 2020*; *Yang and Klee, 2000*). In the case of CHPs, and particularly of CHP3, $Ca^{2+}$ binding to the C-terminal EF-hands (EF-3 in CHP3) affects the conformation also of the N-terminal lobe (*Figure 7*), most likely via the hydrophobic cluster at the interface between N- and C-lobes, which is characteristic for the so-called CPV (calcineurin B, p22/CHP1, visinin) subfamily of EFCaBPs (*Kawasaki and Kretsinger, 2017*). Such conformational changes of CHP might affect the overall structure of the complex formed by CHPs and NHE1. All CHP structures so far, the recent cryo-EM structure of the NHE1:CHP1 complex (*Dong et al., 2021*), and structures of CHP1 (*Mishima et al., 2007*) and CHP2 (*Ammar et al., 2006*) in complex with the target peptide CBD obtained with NMR and X-ray crystallography, showed one conformation of CHPs with $Ca^{2+}$ (or the $Ca^{2+}$-mimic $Y^{3+}$) and the target peptide bound. The structure of CHP3 has not been determined so far and moreover, structural information of target-free CHPs in the open and closed conformations is lacking. Therefore, further structural studies are required to better understand the molecular mechanisms for NHE1 regulation by $Ca^{2+}$ binding to CHPs. This information would be valuable also to elucidate signal integration through the regulatory domain of NHE1, which acts as signalling hub interacting with several regulatory proteins including the $Ca^{2+}$-binding proteins calmodulin and calcineurin (*Pedersen and Counillon, 2019*).

Regarding the lipidation modification, many $Ca^{2+}$ sensor proteins have an N-terminal myristoylation site and often $Ca^{2+}$ binding to the myristoylated EFCaBP causes an exposure of the myristoyl moiety. A surface-exposed myristoyl moiety strongly increases the binding of the $Ca^{2+}$-bound EFCaBP to lipid membranes. This regulatory mechanism is called $Ca^{2+}$-myristoyl switch and was described for a subgroup of neuronal calcium sensor (NCS) proteins (recoverin, hippocalcin, visinin-like proteins, and neurocalcin δ) (*Ames et al., 1997*; *Burgoyne, 2004*). In other NCSs, the position of the myristoyl moiety is not affected by calcium binding, but it is constantly exposed (NCS1, KChIP1) (*McFerran et al., 1999*; *O'Callaghan et al., 2003*) or hidden inside the protein (GCAP1, GCAP2) (*Lim et al., 2014*; *Burgoyne, 2007*; *Burgoyne, 2004*). So far, there is no apparent correlation between number and position of active EF-hands with the position of the myristoyl moiety in NCS proteins. Even for highly homologous myristoylated NCS proteins with known structures, the location of the myristoyl

moiety varies (*Lim et al., 2014*). Thus, structural information and models are not sufficient to assign a calcium-myristoyl switch. For CHP3, a $Ca^{2+}$-myristoyl switch was proposed as mutational analysis indicated that deficiency in $Ca^{2+}$ binding and/or myristoylation of CHP3 reduced the surface expression and activity of NHE1 in the same degree (*Zaun et al., 2012*). Yet, a direct evidence for a $Ca^{2+}$-myristoyl switch in CHP3 was lacking. Our data indicate an independent regulation of CHP3 by $Ca^{2+}$ binding and myristoylation, and contradict the presence of a $Ca^{2+}$-myristoyl switch. Firstly, CHP3 myristoylation only slightly increased the hydrophobicity of the $Ca^{2+}$-bound state in comparison to the prominent hydrophobicity increase of recoverin that has a classical $Ca^{2+}$-myristoyl switch (*Ames et al., 1997*). Secondly, myristoylation did not affect $Ca^{2+}$-binding properties of CHP3. Next, myristoylation reduced the thermal stability of free and target-bound CHP3 in a similar manner for both $Ca^{2+}$- and $Mg^{2+}$-bound states. Furthermore, mCHP3 affinities to CBD in both ion-bound states were decreased twofold compared to CHP3. We also provided further evidence for the lack of a $Ca^{2+}$-myristoyl switch in CHP3 by analysing its interaction with liposomes. Although $Ca^{2+}$ addition strongly increased the CHP3 association with POPC/POPS liposomes, the effect was the same for both myristoylated and non-myristoylated proteins. In the closed conformation, the myristic group is usually buried in the protein core of EFCaBPs, however the binding interface differs a lot in different EFCaBPs (*Lim et al., 2014*). We assume that CHP3 resembles GCAPs, which have the myristoyl group enclosed in the protein hydrophobic core independently of $Ca^{2+}$ binding (*Lim et al., 2014*). The membrane binding of CHP3 independent of its myristoylation suggests that different protein regions, for example clusters of hydrophobic or positively charged residues, are involved in proteinlipid interaction. Binding of the target peptide CBD drastically reduced the interaction of CHP3 with lipid membranes that is apparently caused by large structural rearrangements. The CHP3:CBD complex did not bind to liposomes in the presence of $Ca^{2+}$. One can speculate that residues involved in membrane binding of the target-free $Ca^{2+}$-bound CHP3 can be either covered by the target peptide or become inaccessible due to structural rearrangements upon complex formation. In stark contrast to the target-free CHP3, myristoylation increased fourfold the amount of CHP3:CBD complex associated with liposomes. We propose that the target peptide competes with the myristoyl moiety for binding in the hydrophobic pocket and by this way shifts the conformational equilibrium of CHP3, resulting in the exposure of the myristoyl moiety to the environment (*Figure 8*). The competition for the hydrophobic pocket is also in line with the lower affinity of mCHP3 to CBD in comparison to non-myristoylated protein as shown by MST analysis (*Figure 5*). As the effect on affinity is small, the myristoyl moiety binds most likely weakly or transiently to the target-binding pocket. This resembles the dynamic regulatory interaction of disordered protein regions of p53 to its DNA-binding domain (*Krois et al., 2018*) and of Radical-Induced Cell Death 1 (RCD1) to the ligand-binding pocket (*Staby et al., 2021*).

In general, the surface exposure of the myristoyl moiety in myristoylated regulatory proteins can be induced by $Ca^{2+}$ or by small molecule binding, phosphorylation, and pH shift. Respective regulatory mechanisms were termed $Ca^{2+}$-myristoyl switch (neuronal calcium sensor proteins) (*Ames et al., 1997*; *Burgoyne, 2004*), GTP-myristoyl switch (ARF1 GTPase [*Goldberg, 1998*]), myristoyl/phosphoserine switch (*Braun et al., 2000*; *Gaffarogullari et al., 2011*), and pH-dependent myristoyl switch (hisacto-philin [*Hanakam et al., 1996*]). The myristoyl switch induced by proteinprotein interaction described here for CHP3 is a novel mechanism that regulates the association of myristoylated proteins with lipid

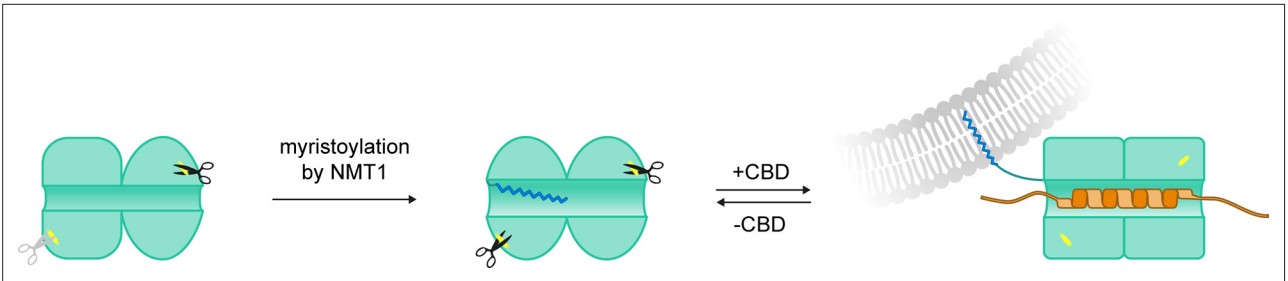

**Figure 8.** Target-myristoyl switch in CHP3. N-terminal myristoylation of CHP3 by NMT1 increases the local flexibility in the N-lobe, as it accelerated its tryptic cleavage. The myristoyl moiety most likely binds weakly or transiently to the target-binding pocket in both $Ca^{2+}$- and $Mg^{2+}$-bound states, as myristoylation reduced twofold the affinity of CHP3 to CBD. CBD binding causes displacement from the hydrophobic pocket and the surface exposure of the myristoyl moiety resulting in enhanced CHP3 binding to lipid membranes in both $Ca^{2+}$- and $Mg^{2+}$-bound states.

membranes. We propose to call it 'target-myristoyl switch' (**Figure 8**). This mechanism was supported by LC–MS/MS analyses, which exclusively detected the N-terminally myristoylated CHP3 in brain membrane fraction (**Figure 6C**) and in a preparation from platelets containing integral membrane proteins and lipid-anchored proteins (**Figure 6—figure supplement 1**). We suggest that the target-myristoyl switch is the canonical mechanism to anchor the regulatory domain of NHE1 to the plasma membrane, which appears to be the molecular basis for an increase of NHE1 surface stability by CHP3. A defined conformation enforced by the membrane anchor may also regulate NHE1 activity. As the second mode of regulation, $Ca^{2+}$ binding to CHP3 can affect NHE1:CHP3 interaction in two ways. $Ca^{2+}$ directly increases the binding affinity of CHP3 for NHE1, and indirectly increases the probability to interact with NHE1 via enhanced association of free CHP3 with lipid membranes. Our hypothesis is in line with the previous observation that both, CHP3 myristoylation and $Ca^{2+}$ binding are important for NHE1 surface stability when both proteins were co-expressed in AP-1 cell line (**Zaun et al., 2012**).

It will be interesting to investigate whether the interaction of CHP3 with other described target proteins, such as with subunit 4 of COP9 signalosome, GSK3ß, and calcineurin A (**Gutierrez-Ford et al., 2003**; **Levay and Slepak, 2014**; **Kobayashi et al., 2015**) can induce the target-myristoyl switch and how $Ca^{2+}$ binding affects these interactions. Structural characterization of CHP3 in different conformations is needed to further explore the underlying molecular basis of the regulatory mechanism. The target-myristoyl switch might exist in other proteins including CHP1 and CHP2 and can be explored by comparison of the membrane association for myristoylated and non-myristoylated complexes.

CHP3 is an emerging important player in the cellular $Ca^{2+}$ signalling network that is involved in regulation of cancerogenesis, cardiac hypertrophy, and neuronal development. Our data can help to better understand the molecular mechanisms of regulation of these processes by CHP3 and of signal integration by NHE1.

## Materials and methods

**Key resources table**

| Reagent type (species) or resource | Designation | Source or reference | Identifiers | Additional information |
|---|---|---|---|---|
| Recombinant DNA reagent | pETDuet-1Δ6His_hNMT_Nef | DOI:10.1371/journal.pone.0010081 | | Provided by Prof. Willbold (Forschungszentrum Jülich, Germany) |
| Peptide, recombinant protein | Bovine serum albumin (BSA) | Sigma-Aldrich | Cat.# A7906 | For production of mCHP3 and mRec |
| Peptide, recombinant protein | Trypsin | Sigma-Aldrich | Cat.# T1426 | For limited trypsinolysis |
| Chemical compound, drug | Myristic acid | Carl Roth GmbH | Cat.# 6469.1 | For production of mCHP3 and mRec |
| Chemical compound, drug | POPC | Avanti Polar Lipids, Inc | Cat.# 850457C | Liposome preparation |
| Chemical compound, drug | POPS | Avanti Polar Lipids, Inc | Cat.# 840034C | Liposome preparation |
| Chemical compound, drug | ProteOrange | Lumiprobe | Cat.# 40210 | Dye for FPH assay |
| Commercial assay, kit | Monolith Series Protein labeling kit – RED-NHS | Nanotemper Technologies | Cat.# MO-001 | MST labeling kit |
| Commercial assay, kit | Monolith Series Premium Capillaries | Nanotemper Technologies | Cat.# MO-K025 | MST capillaries |
| Software, algorithm | Data2Dynamics Software | https://github.com/Data2Dynamics/d2d; **Kreutz, 2023** | | For statistical analysis of datasets consisted of technical and biological replicates |
| Other | 16% Novex Tricine Gel | Thermo Fisher Scientific | Cat.# EC66955BOX | SDS–PAGE system for separation of CHP3 and mCHP3 |
| Other | 96-well black plates | Greiner Bio-One | Cat.# 655076 | For FPH assay |

## Plasmids

The generation of all plasmids was performed using NEBuilder HiFi DNA assembly Master Mix (New England Biolabs GmbH, Frankfurt, Germany). Protein-coding sequences (CDS) were inserted without any additions for affinity tags if not stated otherwise. The pETDuet-1Δ6His_hNMT_Nef containing the codon optimized CDS of the human N-myristoyltransferase 1 (NMT1, Gene ID 4836, UniProtKB ID P30419) was kindly provided by Prof. Willbold (Forschungszentrum Jülich, Germany). To obtain the pETDuet1_NMT1_TESC plasmid, the codon optimized (*E. coli*) CDS of human CHP3 (TESC, Gene ID 54997, UniProtKB ID Q96BS2) was cloned into the second multiple cloning site of the pETDuet-1Δ6His_hNMT_Nef plasmid replacing the Nef CDS. The NMT1 CDS was deleted from the pETDuet1_NMT1_TESC plasmid to generate the plasmid pETDuet1_TESC. For co-expression of the NHE1 CBD:CHP3 complexes, the codon optimized (*E. coli*) CDS of a minimal NHE1 CBD with three additional histidines added C-terminally (NHE1 residues 525–545, MRSINEEIHTQFLDHLLTGIEDICGHYG HHHHHH, CBDHis) was used. In case of the pETDuet1_CBDHis_TESC plasmid, the CBDHis CDS was inserted in the first multiple cloning site, whereas for the pETDuet1_NMT1_CBDHis_TESC plasmid, the CBDHis CDS, and the linker region containing T7 promoter/lac operator and ribosome-binding site was inserted between NMT1 and CHP3 CDSs. The plasmid pETDuet1_Rec containing the codon optimized CDS of human recoverin (Gene ID 5957, UniProtKB ID P35243) in the second multiple cloning site was obtained from BioCat (Heidelberg, Germany). NMT1 CDS was inserted into the first site of this plasmid to obtain the pETDuet1_NMT1_Rec plasmid.

## Protein productions and purifications

We adapted the protocol of *Glück et al., 2010* for the production of recombinant myristoylated CHP3 (mCHP3) by co-expression with human NMT1. Pre-cultures in TB$^{carb}$ medium were made from a single colony of *E. coli* BL21 (DE3) transformed with the pETDuet1_NMT1_TESC plasmid. Main cultures in TB$^{carb}$ medium containing the surfactant Antifoam 204 (Merck KGaA, Darmstadt, Germany) were inoculated to a final $OD_{600} = 0.1$ and incubated at 37°C. At $OD_{600} = 0.6$, the temperature was shifted to 28°C and 3% ethanol (vol/vol) was added. Myristic acid (mixed with 4% bovine serum albumin) was added at $OD_{600} = 0.70–0.75$. The protein production was induced at $OD_{600} = 0.8$ with 0.5 mM IPTG (isopropyl β-D-1-thiogalactopyranoside). After 4-hr incubation, cells were harvested and stored at −80°C. Non-myristoylated CHP3 was produced similarly using the plasmid without NMT1 (pETDuet1_TESC); no antifoam, ethanol and myristic acid were added to the medium. The expression was performed at 30°C overnight.

All chromatography columns were obtained from Cytiva (Marlborough, MS, USA). CHP3 and mCHP3 were purified using Ca$^{2+}$-dependent HIC based on protocols established for other EFCaBPs (*Tanaka et al., 1984*; *Wei and Lee, 1997*). Briefly, sedimented cells were resuspended in buffer A (20 mM HEPES (4-(2-hydroxyethyl)piperazine-1-ethanesulfonic acid), 150 mM NaCl, 10 mM DTT (dithiothreitol), 10% glycerol [wt/vol], pH 7.2) containing 10 mM CaCl$_2$ and 0.3 mM Pefabloc (Carl Roth GmbH, Karlsruhe, Germany), and lysed with a cell disruptor (Constant systems, Daventry, UK). Cell debris was removed by centrifugation and the supernatant was loaded onto a HiPrep 16/10 Phenyl HP column equilibrated with buffer A containing 10 mM CaCl$_2$. After removal of unbound material with 10 column volumes of the equilibration buffer, the protein was eluted with buffer A containing 10 mM EGTA (ethylene glycol bis(2-aminoethyl ether)-N,N,N′,N′-tetraacetic acid), and the eluate was mixed with an equal volume of buffer A containing 10 mM MgCl$_2$. The eluted protein was concentrated and loaded onto a pre-equilibrated HiLoad 26/600 Superdex 75 column. Buffer A containing 10 mM MgCl$_2$ was used as a running buffer. Fractions containing the purified protein were pooled, concentrated, mixed 1:1 with the cryo-storage buffer (2 M sucrose in gel filtration buffer), flash frozen in liquid nitrogen, and stored at −80°C.

Recoverin and mRecoverin were produced and purified as described above for CHP3 and mCHP3 using the pETDuet1_Rec and pETDuet1_NMT1_Rec plasmids, respectively. The fusion construct of maltose-binding protein and NHE1 CBD (MBP-CBD) was produced and purified as described previously (*Fuchs et al., 2018*).

The production of CHP3:CBD and mCHP3:CBD complexes was performed as described above using pETDuet1_CBDHis_TESC or pETDuet1_NMT1_CBDHis_TESC plasmids, respectively. Cells were lysed in the presence of 20 mM imidazol. The complexes were purified by IMAC followed by gel filtration. Cell lysate was loaded onto a HisTrap FF 5 ml column pre-equilibrated with buffer A containing

10 mM CaCl$_2$ and 20 mM imidazol. After removal of unbound material with 10 column volumes of the equilibration buffer, elution was done with a linear gradient from 20 to 600 mM imidazol over 20 column volumes. For gel filtration, buffer A containing 10 mM CaCl$_2$ was used as a running buffer.

## Native mass spectrometry

The degree of myristoylation was determined by native MS. For native ion mobility separation (IM)-MS analysis, proteins were transferred into 200 mM ammonium acetate, pH 6.7. Capillaries for nanoflow ESI were prepared in-house from borosilicate glass capillaries 1.0 mm OD × 0.78 mm ID (Harvard Apparatus, Holliston, MA, USA) using a micropipette puller Model P-97 (Sutter Instruments, Novato, CA, USA) and coated with gold using a sputter coater 108 auto (Cressington TESCAN GmbH, Dortmund, Germany). 3–5 µl of 20 µM protein were loaded per experiment. Measurements were carried out with quadrupole-IM-MS-ToF instrument Synapt G1 HDMS (Waters Corporation, Milford, MA, USA and MS Vision, Almere, The Netherlands) equipped with a 32,000 $m/z$ range quadrupole. Pressures were 8 mbar backing in the source, 0.017 mbar argon in the trap and transfer collision cell and 0.5 mbar nitrogen in the ion mobility cell. Instrument parameters were as follow: capillary voltage 1.2 kV, source temperature 60°C, sample cone voltage 100 V, extraction cone voltage 10 V, acceleration voltages in the trap and transfer cell 5 and 35 V, respectively, IM bias potential 22 V, wave velocity and height 300 m/s and 8 V. Spectra were acquired in the positive ion mode in the $m/z$ range 1000–6000. Spectra recorded for cesium iodide at 50 mg/ml in 50% (vol/vol) isopropanol were used for external mass calibration. Data were processed and analysed using MassLynx Software version 4.1 (Waters Corporation) and UniDec version 2.7.3 (*Marty et al., 2015*). For charge state deconvolution, smoothing and subtraction of linear background from raw spectra were done.

## MS sequencing of native CHP3

Mouse brain membranes were prepared, solubilized with ComplexioLyte 47 (Logopharm, Germany) and protein complexes were separated by BN-PAGE as described (*Schwenk et al., 2012*). A section corresponding to an apparent molecular weight of 630 kDa (the migration size of native NHE1 complexes) was excised and subjected to MS analysis (as described below). Mouse platelets were prepared from fresh blood by differential centrifugation and 20 µg washed with 50 µl 100 mM Na$_2$CO$_3$ pH 11 (10 min at 4°C, followed by ultracentrifugation at 233,000 × $g$ for 20 min). The resulting pellet was dissolved in Laemmli buffer and separated on an SDS–PAGE gel and silver stained. The section <50 kDa was excised, in-gel digested with trypsin, the peptides were dissolved in 20 µl 0.5% (vol/vol) trifluoroacetic acid, and 1 µl was analysed on an LC–MS/MS setup (UltiMate 3000 RSLCnano HPLC/QExactive HF-X mass spectrometer, both Thermo Scientific, Germany) as described (*Kollewe et al., 2021*). MS/MS data were extracted and searched against the UniProtKB/Swiss-Prot database (mouse, rat, human, release 20220525; Mascot search engine version 2.7, Matrix Science, UK) with the following settings: Acetyl (Protein N-term), Carbamidomethyl (C), Gln->pyro-Glu (N-term Q), Glu->pyro-Glu (N-term E), Oxidation (M), Propionamide (C) as variable modifications, one missed cleavage allowed, with or without myristoylation (N-term G) or in error-tolerant mode; precursor/fragment mass tolerance was ±10 ppm/±25 mmu, significance threshold p < 0.05.

## Fluorescence-based assays

The conformational changes of the protein in response to ion binding were probed with the FPH assay (*Liang et al., 2020*) and by monitoring the intrinsic tryptophan fluorescence. The buffer of the protein solution was exchanged to the assay buffer (20 mM HEPES, 150 mM NaCl, 1 mM TCEP (Tris(2-carboxyethyl)phosphin), 2 mM MgCl$_2$, pH 7.2) containing 1 mM EGTA using Zeba spin desalting columns (Thermo Fisher Scientific Inc, Waltham, MA, USA). The FPH assay was modified as follows. 500 µl of 1.5 µM protein and 5 µM ProteOrange (Lumiprobe GmbH, Hannover, Germany) in the assay buffer were mixed and incubated for 30 min in the dark. The fluorescence at 585 nm (excitation at 470 nm) was measured continuously with the Cary Eclipse spectrofluorimeter (Agilent Technologies, Santa Clara, CA, USA). Every 60 s a new compound was added in the following order: 2 mM CaCl$_2$, 3 mM EGTA, 3 mM EDTA (ethylenediaminetetraacetic acid), and 4 mM CaCl$_2$.

Determination of Ca$^{2+}$ EC50 values was performed using the FPH assay, mixing 1.5 µM protein and 5 µM ProteOrange with varied (0.34 µM to 8.0 mM) concentrations of CaCl$_2$. The assay was carried out in 96-well black plates (Greiner Bio-One, Frickenhausen, Germany) with a sample volume of 100 µl.

For each biological replicate, three to four technical replicates were measured. The fluorescence at 590 nm with excitation at 485 nm ($F_{590}$) was measured with the BioTek Synergy 2 plate reader (Agilent Technologies, Santa Clara, CA, USA). The data were fitted using the Hill equation

$$F_{590}\left(c\left(Ca^{2+}\right)\right) = F_0 + \frac{F_{max}}{1 + \left(\dfrac{EC50}{c\left(Ca^{2+}\right)}\right)^n}$$

to estimate parameters from data which has the a priori unknown parameters: $F_0$ ($F_{590}$ in the absence of $Ca^{2+}$), $F_{max}$ (maximal change of $F_{590}$), EC50, and $n$ (Hill coefficient). The parameters EC50 and the Hill coefficient $n$ represent biological quantities that are global and therefore shared between the three biological replicates. The parameters $F_0$ and $F_{max}$ are affected by experimental procedures and therefore different between the three biological replicates. Adding one parameter for dose-independent measurement noise, a total of nine parameters has to be estimated simultaneously for each condition. Parameter uncertainty can be assessed using the profile likelihood method, which is also referred to as error surface plots. We utilized optimization to obtain the best parameter set that minimizes deviation between model and data in terms of the negative log-likelihood. Uncertainty of the parameters is quantified using profile likelihood method (*Kreutz et al., 2013*), which is also termed error surface plots in the literature. This fitting procedure was independently applied to both conditions in order to obtain EC50 values and corresponding uncertainties for both conditions. Data2Dynamics Software (*Raue et al., 2015*) used for EC50 determination from data including biological and technical replicates is available at GitHub (https://github.com/Data2Dynamics/d2d, copy archived at *Kreutz, 2023*).

For monitoring the intrinsic tryptophan fluorescence, 500 µl of 2.5 µM of the respective protein in assay buffer were prepared. Fluorescence was measured at 330 nm with an excitation at 280 nm. The time traces were recorded as described above for the FPH assay. For the tryptophan fluorescence spectra, a protein concentration of 5 µM was used and emission spectra of CHP3 and mCHP3 were measured from 300 to 380 nm (excitation at 280 nm) in the presence of 2 mM $Mg^{2+}$ plus 1 mM EGTA and with further addition of 2 mM $Ca^{2+}$.

All assays were performed with three individually purified biological replicates of CHP3 and mCHP3.

## Thermal stability analysis with nanoDSF

The thermal stability of CHP3, mCHP3, and the complexes with CBD was compared at different conditions using nano differential scanning fluorimetry (nanoDSF). Buffer of the protein solutions was exchanged to the assay buffer as above; the final concentration was adjusted to 100 µM and thermal stability was measured in the presence of 10 mM of either $MgCl_2$, $CaCl_2$, $MgCl_2$ and $CaCl_2$, or EDTA. Each sample was measured with two technical and three independent biological replicates in standard capillaries (Nanotemper Technologies, Munich, Germany) with the Prometheus NT.48 device (Nanotemper Technologies). Measurements were carried out from 20 to 95°C with a heating rate of 1°C/min. Unfolding of proteins analysed was irreversible, thus apparent melting temperatures ($T_m^{app}$) are given. They were determined as an inflection point of the unfolding curves using the PR.ThermControl software (Nanotemper Technologies).

## Limited proteolysis

Next, we probed the proteolytic stability of CHP3, mCHP3, and of their complexes with CBD. Buffer of the protein solutions was exchanged to the assay buffer containing either 2 mM $MgCl_2$ + 0.5 mM EGTA, 2 mM $CaCl_2$, 2 mM $MgCl_2$ + 2 mM $CaCl_2$, or 2 mM EDTA as an additive. The final protein concentration was adjusted to 0.5 mg/ml for each sample. Each reaction was started with the addition of 0.01 mg/ml trypsin (Sigma Aldrich, Cat. No. T1426) dissolved in 1 mM HCl, and performed for 1 hr on ice. Samples were taken after 5, 15, 30, 45, and 60 min, directly mixed with an equal volume of 2× reducing NuPAGE LDS sample buffer (Thermo Fisher Scientific) and boiled for 5 min at 95°C. The proteolytic fragments were separated on 12% NuPAGE Bis-Tris gels (Thermo Fisher Scientific).

To determine which trypsin cleavage sites in CHP3 are accessible for proteolysis, the major proteolytic fragments were analysed by ESI MS. 100 µl of CHP3 and mCHP3 in the assay buffer with 2 mM $CaCl_2$ and 2 mM $MgCl_2$ were incubated with trypsin for 15 min. Protein fragments were separated on 16% Novex Tricine gels (Thermo Fisher Scientific). The major bands were cut, minced into smaller pieces and loaded on top of 300 kDa Nanosep centrifugal devices (Pall Corporation, Port Washington,

NY, USA). Proteins were eluted two times with 400 µl of the assay buffer containing 0.1% SDS by centrifugation at 14,000 × $g$ for 20 min. Eluates were concentrated with Vivaspin500 5,000 MWCO concentrators (Sartorius AG, Goettingen, Germany). SDS was removed by precipitation with 0.2 M KCl followed by 10 min centrifugation at 14,000 × $g$. 10 µl of each sample were analysed with liquid chromatography (Thermo Fisher Ultimate 3000 with Phenomenex Jupiter 5 µm C4 300 Å 50 × 2 mm) coupled to high-resolution ESI MS (Maxis 4G, Bruker Corporation, Billerica, MA, USA). The resulting peptide masses were compared to all possible trypsin digestion fragments of CHP3 or mCHP3 to identify the cleavage sites.

## Microscale thermophoresis

MST was performed using the Monolith Pico device (Nanotemper Technologies) as described previously (*Fuchs et al., 2018*). The only difference was the use of the measuring buffer containing either 10 mM CaCl$_2$ or 10 mM MgCl$_2$ (20 mM HEPES, 5 mM TCEP, 150 mM NaCl, 0.05% Tween-20, 10 mM MgCl$_2$, or 10 mM CaCl$_2$; pH 7.2). All measurements were performed with three independently produced and purified protein replicates of CHP3, mCHP3, and MBP-CBD. Every condition was titrated and measured five times per biological replicate (technical replicates).

Mass action kinetics of a one-site binding model was utilized to derive the fitting function for dissociation constant ($K_D$) determination: Let $A$ and $B^*$ denote the concentrations of unlabelled and labelled binding partners respectively, and the concentration of the protein complex by $AB^*$. The total amounts of $A$ and $B^*$ ($A_{tot}$ and $B^*_{tot}$) can be controlled by the experimental setup. In our application, $B^*_{tot}$ =10 nM was fixed and $A_{tot}$ =[MBP-CBD] was the independent variable for the dose response analysis. The thermophoresis data were normalized and fitted with global non-linear regression to the dose response function (*Scheuermann et al., 2016*):

$$F_n\left(A_{tot}\right) = \left(\frac{B^*}{B^*_{tot}} \cdot F_{n,\,B^*}\right) + \left(\frac{AB^*}{B^*_{tot}} \cdot F_{n,AB^*}\right)$$

with $F_n\left(A_{tot}\right)$ – thermophoresis (change of normalized fluorescence) as a function of MBP-CBD concentration, $F_{n,\,B^*}$ and $F_{n,AB^*}$ – thermophoresis of the labelled partner and the complex, respectively. The concentration of the complex is given by

$$AB^* = \frac{\left(K_D + A_{tot} + B^*_{tot}\right) - \sqrt{\left(K_D + A_{tot} + B^*_{tot}\right)^2 - 4 \cdot A_{tot} \cdot B^*_{tot}}}{2}$$

which follows from solving a quadratic function originating from mass action kinetics combined with equilibrium assumption for the signal strength.

Simultaneously, all individual parameters ($B^*$, $F_{n,\,B^*}$, $F_{n,AB^*}$) per biological replicate were fitted together with the global $K_D$ value. To estimate the noise level within the technical replicates we used an error model with absolute and relative errors represented by two parameters per biological replicate. With three biological replicates, this results in a total of 16 parameters that were fitted simultaneously. We utilized deterministic multistart optimization to obtain the best parameter set that minimizes deviation between model and data in terms of the negative log-likelihood. Uncertainty of the parameter of interest $K_D$ is quantified using profile likelihood method (*Kreutz et al., 2013*), which is also termed error surface plots in the literature. This fitting procedure was independently applied to each condition in order to obtain $K_D$ values and corresponding uncertainties for all four conditions. Data2Dynamics Software (*Raue et al., 2015*) used for $K_D$ determination from data including biological and technical replicates is available at GitHub (https://github.com/Data2Dynamics/d2d, copy archived at *Kreutz, 2023*).

## Interaction of proteins with lipid membranes

The POPC:POPS (Avanti Polar Lipids, Birmingham, AL, USA) mixture (3:1 molar ratio) in chloroform was dried in round bottom glass vials under a nitrogen stream. The dried lipid film was resuspended in liposome buffer (20 mM HEPES, 150 mM NaCl, 10% sucrose, pH 7.2) at a final concentration of 4 mM, sonified three times for 20 s with 40 s breaks (Bandelin Sonorex ultrasonic bath). Then liposomes were subjected to 10 freeze–thaw cycles and extruded through a 1-µm polypropylene membrane (Avestin,

Ottawa, Canada). The size of the resulting liposomes was controlled with DLS (DynaPro NanoStar, Wyatt Technology, Santa Barbara, CA, USA). Liposomes were stored at room temperature in the dark.

The lipidprotein co-sedimentation assay was performed mostly as described previously (*Zozulya and Stryer, 1992*). Briefly, proteins were transferred to the assay buffer. Liposomes were washed once with the assay buffer containing either 2 mM $CaCl_2$ ($Ca^{2+}$ condition) or 1 mM EGTA ($Mg^{2+}$ condition) and mixed with the protein in the respective buffer. Final lipid and protein concentrations were 1.4 mM and 10 µM, respectively. The samples were incubated for 20 min at 24°C with constant mixing and centrifuged at 17,000 × *g* for 15 min. The pellet was washed once with the respective buffer and resuspended in 1× reducing NuPAGE LDS sample buffer (Thermo Fisher Scientific). Samples were analysed using 4-12% NuPAGE Bis-Tris gels (Thermo Fisher Scientific).

The lipid-binding experiments were performed with three individually purified biological replicates of CHP3 and mCHP3 and respective complexes with CBD. Three technical replicates were analysed for Recoverin and mRecoverin as controls. Densitometry of protein bands was done with ImageJ (NIH). Band intensities were normalized to the $Mg^{2+}$ condition of the respective non-myristoylated protein.

## Acknowledgements

This work was supported by the Deutsche Forschungsgemeinschaft (DFG, German Research Foundation) under Germany's Excellence Strategy (CIBSS – EXC-2189 – Project ID 390939984) in the form of project funding to BW, BF, CH, CK and of CIBSS Launchpad Program Funds to EM; by the DFG through Project-ID 403222702/SFB 1381 to CH, US, and BW, through Project-ID 431984000/SFB 1453 to BF and CH, and through project-ID 278002225/RTG 2202 to CH; and by the German Ministry of Education and Research by grant EA:Sys (FKZ031L0080) to CK. We thank Prof. Willbold (Forschungszentrum Jülich, Germany) for kindly providing the pETDuet-1Δ6His_hNMT_Nef plasmid, Dr Michael Pfeffer (MS Service at University of Basel) for performing ESI mass spectrometry, and Michal Rössler (CIBSS, University of Freiburg) for supporting the artwork in *Figures 7–8*.

## Additional information

### Competing interests

Uwe Schulte: is an employee and shareholder of Logopharm GmbH that produces ComplexioLyte 47 used in this study. Bernd Fakler: is shareholder of Logopharm GmbH. The company provides ComplexioLyte reagents to academic institutions on a non-profit basis. The other authors declare that no competing interests exist.

### Funding

| Funder | Grant reference number | Author |
| --- | --- | --- |
| Deutsche Forschungsgemeinschaft | EXC-2189 - 390939984 | Clemens Kreutz<br>Bettina Warscheid<br>Carola Hunte<br>Bernd Fakler<br>Evgeny V Mymrikov |
| Deutsche Forschungsgemeinschaft | SFB 1381 - 403222702 | Uwe Schulte<br>Carola Hunte<br>Bettina Warscheid |
| Deutsche Forschungsgemeinschaft | SFB 1453 - 431984000 | Bernd Fakler<br>Carola Hunte |
| Deutsche Forschungsgemeinschaft | RTG 2202 - 278002225 | Carola Hunte |
| Bundesministerium für Bildung und Forschung | FKZ031L0080 | Clemens Kreutz |

The funders had no role in study design, data collection, and interpretation, or the decision to submit the work for publication.

## Author contributions
Florian Becker, Conceptualization, Formal analysis, Validation, Investigation, Visualization, Methodology, Writing – original draft, Writing – review and editing; Simon Fuchs, Conceptualization, Formal analysis, Investigation, Visualization, Writing – original draft; Lukas Refisch, Software, Formal analysis, Validation, Investigation, Visualization, Methodology; Friedel Drepper, Uwe Schulte, Formal analysis, Validation, Investigation, Visualization, Methodology; Wolfgang Bildl, Shuo Liang, Jonas Immanuel Heinicke, Sierra C Hansen, Investigation; Clemens Kreutz, Software, Formal analysis, Supervision, Validation, Methodology; Bettina Warscheid, Bernd Fakler, Supervision, Validation, Methodology; Evgeny V Mymrikov, Conceptualization, Data curation, Formal analysis, Supervision, Validation, Investigation, Visualization, Methodology, Writing – original draft, Writing – review and editing; Carola Hunte, Conceptualization, Data curation, Formal analysis, Supervision, Funding acquisition, Validation, Writing – original draft, Project administration, Writing – review and editing

## Author ORCIDs
Florian Becker ![ORCID] http://orcid.org/0000-0002-5889-8728
Lukas Refisch ![ORCID] http://orcid.org/0000-0002-9049-1944
Friedel Drepper ![ORCID] http://orcid.org/0000-0002-2043-5795
Uwe Schulte ![ORCID] http://orcid.org/0000-0003-3557-0591
Bettina Warscheid ![ORCID] http://orcid.org/0000-0001-5096-1975
Evgeny V Mymrikov ![ORCID] http://orcid.org/0000-0003-4817-6278
Carola Hunte ![ORCID] http://orcid.org/0000-0002-0826-3986

## Decision letter and Author response
Decision letter https://doi.org/10.7554/eLife.83868.sa1
Author response https://doi.org/10.7554/eLife.83868.sa2

---

# Additional files

## Supplementary files
- MDAR checklist

## Data availability
All data generated or analyzed during this study are included in the manuscript. Source files have been provided for Figures 1, 4, 6, and Figure 4—figure supplement 1. Data2Dynamics Software used for EC50 and $K_D$ determination is available at GitHub (https://github.com/Data2Dynamics/d2d, copy archived at *Kreutz, 2023*).

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
