## [Editor Report]

In this work, the authors provide important mechanistic insights into how the intracellular effector protein Calcineurin B homologous protein 3 (CHP3) can be regulated in a calcium-independent manner to expose its lipid membrane binding site. Compelling evidence demonstrates a binding partner protein (NHE1) triggers a conformation change and exposure of the myristoyl group in CHP3 resulting in membrane association. This provides mechanistic insight into the signalling mechanisms achieved by CHP3 in a target-binding dependent manner, which will be of broad scientific interest.

---

## [Decision Letter]

**Decision letter after peer review:**

Thank you for submitting your article "Conformational dynamics and target-dependent myristoyl switch of calcineurin B homologous protein 3" for consideration by *eLife*. Your article has been reviewed by 3 peer reviewers, and the evaluation has been overseen by a Reviewing Editor and Richard Aldrich as the Senior Editor. The reviewers have opted to remain anonymous.

Essential revisions:

The manuscript has been improved but there are some remaining issues that need to be addressed, as outlined below:

In particular, please address the implications of Ca^2+^ binding affinities of the CHP proteins to the physiological concentrations of Ca^2+^ (reviewer #1). Enhanced descriptions of the state of the protein would be helpful (see reviewer #3). Reviewer 2 provides many suggestions for improvement to the submitted manuscript, many of which I think can be easily addressed.

*Reviewer #1 (Recommendations for the authors):*

Please address the implications of Ca^2+^ binding affinities of the CHP proteins to the physiological concentrations of Ca^2+^. Please use submicromolar-free Ca^2+^ in all experiments.

Have you considered the effect of the poly His tail of the CBD peptide on binding to CHP3 and ions?

What is the stoichiometry of CBD binding to CHP3?

*Reviewer #2 (Recommendations for the authors):*

Several of the EF-hand calcium-binding proteins are myristoylated and target membranes, and the presence of calcium-binding to induce conformational switches that unleashes a myristoyl chain has been shown in some proteins e.g. recoverin. For others, e.g. NCS1, a lack of a calcium-myristoyl switch seems to be the currently accepted mechanism. The manuscript by Becker and co-workers addresses the important question of how the myristoylated calcium-binding EF-hand protein CHP3 is regulated by ion-binding, and how binding of one of its targets, the sodium exchange NHE1 (or a peptide thereof) impacts membrane binding. The manuscript is very well written, and the data are presented in a logical way. The conclusion of a novel mechanism – a target-dependent myristoyl switch seems largely warranted by the data. There are certain wordings and conclusions that would benefit from a more precise language (or down toning) and some additional controls, and I find the discussion lengthy and repetitive to the introduction and result section and would have wished for a broader discussion putting the data in perspective for the function of NHE1 and other ligand-dependent switches. Addressing the below points would increase the value of the work and the applicability of its finding. In no prioritized order, I suggest the following:

The model and the title "target-dependent" suggest to me that not all targets may induce the switch and that it depends on which target it is bound to. However, only one target has been investigated. Perhaps the more correct term would be target-binding myristoyl switch. And – on that note, how is this mechanism different from the ligand-dependent myristoyl switch previously described and mentioned by the authors? This is not clear from the discussion. Some of the myristoylated calcium-binding EF-hand proteins do not have a calcium-myristoyl switch, e.g. NCS1 and other, are there any correlation between the number of calcium-bound ions and the type of switch? A more thorough discussion on this part is needed.

The claim that the tryptophan moves from a more hydrophobic into solvent exposure cannot be concluded from the intensities alone. The induced dipole in the excited state interacts with the surroundings through dipole-dipole interactions and thus a wavelength shift is expected from hydrophobic (lower wavelength) to solvent (higher wavelength). Please include these data and show the full spectra.

Were the melting curves measured by nanoDSF reversible? If not, these are apparent Tms. Can anything be inferred from the slopes (change in enthalpy) upon unfolding? If the structures are different these should be different too.

For typsinolysis, what is the coverage? Are there equally dispersed lysines/arginines across the sequence or are there silent regions not captured by this technique? It may be one reason why the nanoDSF data is not correlating with the stability measurements. However, a more detailed analysis of the gels in the supplement data and in the main figure, reveals additional bands for the mCHP3 around 15kDa (3 bands instead of 2) and below 10kDa (3 bands instead of 2) that are not present in the CHP3. These bands would be relevant to analyze. I disagree with the conclusion that by binding the peptide, there are substantial structural changes, the data can only show a lack of cleavage, which may be caused by coverage by binding, and stabilization by binding (and here it would be good to correlate to the nanoDSF data). The use of the word conformation dynamics is not appropriate, no kinetics have been measured, use instead conformational states.

The two-fold difference in affinity between CHP3 and mCHP3 for CBD is small, and thus one could argue that this is insignificant. However, it has a resemblance to some of the works on the disordered flanking regions of folded proteins that dynamically occupy a binding site in the folded domain (see e.g. works on RST and p53 with tails PMID: 34687712 and PMID: 30420502) – similar effect were seen. Would be a good addition to the discussion.

The discussion is lengthy and not forward-looking. The entire first paragraph is mostly a repetition of the introduction. It would have been appropriate to include other already mentions points above as well as a reflection of how the other calcium-binding sensitive partners to NHE1 (calmodulin and calcineurin) and the membrane interaction of NHE1 would jointly organize NHE1 at the membrane.

In the discussion, it is concluded that Ca^2+^ induced higher flexibility. How is this conclusion reached? The same pattern of peptide cleavage is reached, and the protein is destabilized, which could be due to a different conformation as also probed with the hydrophobicity assay. It is not necessarily more flexible. Please elaborate on this to make the argumentation clearer or remove it from the discussion if not fully supported. Also, in the discussion, it is suggested that the conformation of the N-terminal lobe is affected, which data supports this? How can this be inferred from the presented data? It is also indicated in Figure 7.

The change in the amount of CHP3 in the liposome fraction upon CBD binding is very interesting and suggests that the myristoyl group and the CDB act cooperatively. Are there any relevant exposed residues on the exposed side of the bound CDB that can explain this? Lysine residues or hydrophobic residues? It would be good to have an AlphaFold2 structure of the complex to help address this.

The 3 remaining EF-hands appear not to bind calcium, where are they mutated, and which properties do they then have? Please include.

Figures

Figure 1 – it appears for both proteins that a fraction of dimers elutes as well to the left of the main peak. Please comment.

Figure 4 – relevant to indicate the potential coverage (Ks and Rs in the sequence) and not just the two where you see cleavage. Aldo the additional bands on the mCHP3 should be acknowledged.

*Reviewer #3 (Recommendations for the authors):*

The description of the limited proteolysis experiment would benefit from additional detail. E.g., how many trypsin sites are in CBD? Is the stoichiometry of CBD:CHP3 1:1 or is an excess of CBD added? In the case of the latter, an experiment to determine whether CBD inhibits trypsin may be considered. Admittingly, this is unlikely to be the case, but it would further strengthen the claim of CBD-mediated stabilization.

---

## [Author Response]

Essential revisions:The manuscript has been improved but there are some remaining issues that need to be addressed, as outlined below:In particular, please address the implications of Ca^2+^ binding affinities of the CHP proteins to the physiological concentrations of Ca^2+^ (reviewer #1). Enhanced descriptions of the state of the protein would be helpful (see reviewer #3). Reviewer 2 provides many suggestions for improvement to the submitted manuscript, many of which I think can be easily addressed.

We thank all reviewers for their valuable feedback on our manuscript and addressed all questions and comments as described in the point-by-point reply. Ca^2+^-binding affinities of EFCaBPs measured in vitro give an estimate of the actual affinity at physiological conditions in a cell. A number of factors including competition with Mg^2+^, target binding and other factors can significantly shift Kd to higher or lower values in the cell. We addressed these points specifically in introduction and discussion as described in detail in response to reviewer #1. We also introduced changes throughout the manuscript to consistently describe different states (Ca^2+^-, Mg^2+^- and apo-state) of the protein as well as its conformations (open, closed and target-bound) based on our biochemical characterization. Details of the updates are provided in responses to reviewers #2 and #3. In response to the numerous detailed suggestions provided by reviewer #2, we have implemented updates that enhanced the manuscript (including Trp spectra).

Reviewer #1 (Recommendations for the authors):Please address the implications of Ca^2+^ binding affinities of the CHP proteins to the physiological concentrations of Ca^2+^.

Ca^2+^-binding affinities of EFCaBPs measured in vitro give an estimate of the actual affinity at physiological conditions in a cell. A number of factors including competition with Mg^2+^, target binding and others can significantly shift Kd to higher or lower values. This has been discussed in details by Gifford and colleagues (Ref. 18). Indeed, target binding (NHE1 CBD) strongly increased Ca^2+^-binding affinities for CHP1 and CHP2 (Refs 16-17), whereas the presence of Mg^2+^ decreased the Ca^2+^-binding affinity of CHP3 to 3.5 µM (Ref 4). Assuming that Mg^2+^ would also compete for Ca^2+^ binding to CHP1 and CHP2, the respective affinities for Ca^2+^ could be above the resting Ca^2+^ concentration. At the same time, CHP3 affinity might be impacted by interaction with its target proteins including NHE1, so that the Ca^2+^-binding affinity of the complex could fit to the concentration range of Ca^2+^-signals.

We added the following information to the introduction:

“Further, Ca^2+^-binding affinities of CHP1 and CHP2 strongly increased (45-fold and 42-fold, respectively) upon binding of the NHE1 target peptide (16, 17). This modulation is a common feature in EFCaBPs (18) and should also apply to the CHP3 isoform. CHP3 may thus respond to Ca^2+^ signals with conformational changes when the intracellular Ca^2+^ concentration elevates from 100 nM to 1 µM or higher (19).”

and updated the discussion as follows:

“Ca^2+^-induced conformational changes are a hallmark of Ca^2+^-sensing EFCaBPs, which control the interaction with and the function of specific target proteins (54). Our in vitro experiments show that Ca^2+^ binding induces conformational changes in CHP3, thus it fulfills this prerequisite to respond to Ca^2+^ signals in the cell. Based on the reported affinities of CHP3 for Ca^2+^ and Mg^2+^ (4), CHP3 should bind Mg^2+^ at resting Ca^2+^ concentration of about 0.1 µM in the cell (19, 55) and it could bind Ca^2+^ when intracellular Ca^2+^ concentration rises up to 1 µM during signalling events (55).”

Please use submicromolar-free Ca^2+^ in all experiments.

The use of saturating millimolar Ca^2+^ concentrations and depletion of Ca^2+^ with an excess of EGTA is the commonly used and well-accepted approach to study conformational changes of calcium binding proteins including EFCaBPs (e.g. Zanetti et al. and Krishnakumar (2016); *eLife* 5, DOI: 10.7554/*eLife*.17262; Barret et al. and Marino (2023). Proc Natl Acad Sci U S A 120(15): e2300309120, DOI: 10.1073/pnas.2300309120; Sjogaard-Frich et al. and Pedersen (2021), *eLife* 10, DOI: 10.7554/*eLife*.60889.) We followed this approach, as saturating conditions permit the characterization of uniform conformational states of the protein.

Have you considered the effect of the poly His tail of the CBD peptide on binding to CHP3 and ions?

CBD has naturally three consecutive C-terminal histidine residues which we could exploit for binding properties. We only added 3 more histidine residues, so that a potential interference is minimal.

What is the stoichiometry of CBD binding to CHP3?

The stoichiometry is 1:1. Thank you for pointing this out. We added in the Introduction.

“We previously demonstrated that CHP3 binds at 1:1 ratio to the CHP-binding region of human NHE1 (CBD) with high affinity in the presence of Mg^2+^ (35)”.

Reviewer #2 (Recommendations for the authors):Several of the EF-hand calcium-binding proteins are myristoylated and target membranes, and the presence of calcium-binding to induce conformational switches that unleashes a myristoyl chain has been shown in some proteins e.g. recoverin. For others, e.g. NCS1, a lack of a calcium-myristoyl switch seems to be the currently accepted mechanism. The manuscript by Becker and co-workers addresses the important question of how the myristoylated calcium-binding EF-hand protein CHP3 is regulated by ion-binding, and how binding of one of its targets, the sodium exchange NHE1 (or a peptide thereof) impacts membrane binding. The manuscript is very well written, and the data are presented in a logical way. The conclusion of a novel mechanism – a target-dependent myristoyl switch seems largely warranted by the data. There are certain wordings and conclusions that would benefit from a more precise language (or down toning) and some additional controls, and I find the discussion lengthy and repetitive to the introduction and result section and would have wished for a broader discussion putting the data in perspective for the function of NHE1 and other ligand-dependent switches. Addressing the below points would increase the value of the work and the applicability of its finding. In no prioritized order, I suggest the following:The model and the title "target-dependent" suggest to me that not all targets may induce the switch and that it depends on which target it is bound to. However, only one target has been investigated. Perhaps the more correct term would be target-binding myristoyl switch. And – on that note, how is this mechanism different from the ligand-dependent myristoyl switch previously described and mentioned by the authors? This is not clear from the discussion.

We had named this mechanism target-dependent myristoyl switch to match the terms used in a recent review on protein myristoylation (Ref 25). We agree that target-dependent might be perceived as “that not all targets may induce the switch and that it depends on which target it is bound to.” As we want to emphasize that the target binding elicits the response, we now rename the mechanism to target-myristoyl switch, which is also matching the names of calcium-myristoyl switch (the prototype of the switches) and GTP-myristoyl switch. We introduced the change throughout the manuscript including the title.

“Respective regulatory mechanisms were termed Ca^2+^-myristoyl switch (neuronal calcium sensor proteins) (63, 64), GTP-myristoyl switch (ARF1 GTPase (27)), myristoyl/phosphoserine switch (28, 29), and pH-dependent myristoyl switch (hisactophilin (30)). The myristoyl switch induced by protein-protein interaction described here for CHP3 is a novel mechanism that regulates the association of myristoylated proteins with lipid membranes. We propose to call it “target-myristoyl switch” (Figure 8).”

Some of the myristoylated calcium-binding EF-hand proteins do not have a calcium-myristoyl switch, e.g. NCS1 and other, are there any correlation between the number of calcium-bound ions and the type of switch? A more thorough discussion on this part is needed.

We included the following information in the discussion:

“This regulatory mechanism is called Ca^2+^-myristoyl switch and was described for a subgroup of neuronal calcium sensor (NCS) proteins (recoverin, hippocalcin, visinin-like proteins, and neurocalcin δ) (63, 64). In other NCSs, the position of the myristoyl moiety is not affected by calcium binding, but it is constantly exposed (NCS1, KChIP1) (31, 33) or hidden inside the protein (GCAP1, GCAP2) (23, 54, 64). So far, there is no apparent correlation between number or position of active EF-hands with the position of the myristoyl moiety in NCS proteins. Even for highly homologous myristoylated NCS proteins with known structures, the location of the myristoyl moiety varies (23). Thus, structural information and models are not sufficient to assign a calcium-myristoyl switch.”

The claim that the tryptophan moves from a more hydrophobic into solvent exposure cannot be concluded from the intensities alone. The induced dipole in the excited state interacts with the surroundings through dipole-dipole interactions and thus a wavelength shift is expected from hydrophobic (lower wavelength) to solvent (higher wavelength). Please include these data and show the full spectra.

We appreciate the suggestion and now included the Trp spectra of CHP3 and mCHP3 in Mg^2+^- and Ca^2+^-bound states as Figure 2—figure supplement 2. The spectra show a fluorescence maximum at ~330 nm for CHP3 and mCHP3 in line with the position of Trp191 in the AlphaFold model. It is located in the target binding pocket at the protein surface with the side chain flanked by other hydrophobic residues (Phe187, Ile 190, Ile194, Ile196). Calcium binding did not induce pronounced spectral shifts, but decreased the fluorescence quantum yield. We updated the respective Results section (p.7) as follows:

“According to the 3D model of CHP3 predicted with AlphaFold2.0 (42), the single tryptophan residue (Trp191) is located in CHP3´s hydrophobic target-binding pocket at the protein surface (Figure 2C) in line with the measured emission maxima of CHP3 and mCHP3 of ~330 nm (Figure 2—figure supplement 2). Intensity and emission maximum of intrinsic tryptophan fluorescence depend on the local environment of the residue (43, 44). Ca^2+^ addition decreased the intrinsic fluorescence of CHP3 (Figure 2B) reflecting conformational changes accompanied with changes in the local environment of Trp191 in the hydrophobic pocket. This is in line with an increase of CHP3 hydrophobicity observed with the FPH assay.”

Were the melting curves measured by nanoDSF reversible? If not, these are apparent Tms. Can anything be inferred from the slopes (change in enthalpy) upon unfolding? If the structures are different these should be different too.

Unfolding of free CHP3 and of CHP3 in complex with NHE1 CBD is not reversible. We changed “Tm” for “apparent Tm” (T_m_^app^). We added information in the Materials and methods as follows:

“Unfolding of proteins analysed was irreversible, thus apparent melting temperatures (T_m_^app^) are given. They were determined as an inflection point of the unfolding curves using the PR.ThermControl software (Nanotemper Technologies).”

Due to irreversibility of unfolding, we would not infer any thermodynamic parameters from the unfolding curves.

For typsinolysis, what is the coverage? Are there equally dispersed lysines/arginines across the sequence or are there silent regions not captured by this technique? It may be one reason why the nanoDSF data is not correlating with the stability measurements. However, a more detailed analysis of the gels in the supplement data and in the main figure, reveals additional bands for the mCHP3 around 15kDa (3 bands instead of 2) and below 10kDa (3 bands instead of 2) that are not present in the CHP3. These bands would be relevant to analyze. I disagree with the conclusion that by binding the peptide, there are substantial structural changes, the data can only show a lack of cleavage, which may be caused by coverage by binding, and stabilization by binding (and here it would be good to correlate to the nanoDSF data). The use of the word conformation dynamics is not appropriate, no kinetics have been measured, use instead conformational states.

We integrated information on lysine/arginine position as Figure 4—figure supplement 1A. Cleavage sites are distributed over the entire sequence of CHP3. We added the following sentence in Results:

“Trypsin cleavage sites (Arg and Lys residues) are distributed all over the CHP3 sequence (Figure 4—figure supplement 1A), yet productive cleavage requires not only availability of the site but it requires also flexibility of the peptide chain at the cleavage site to adapt to the active site of the protease (45).”

This update includes addition of reference (Ref 45).

We agree that there are minor differences in the tryptic cleavage of CHP3 and mCHP3. We updated results as follows:

“In addition, a second minor band below fragment 1 appeared for mCHP3 (Figure 4B). The small differences in the tryptic cleavage of CHP3 and mCHP3 indicate a local destabilizing effect of the myristoyl moiety, in line with the lower thermal stability of mCHP3 compared to CHP3.”

Regarding coverage and stabilization by binding: CBD is highly unlikely to inhibit the cleavage by covering the two main cleavage sites, as they are facing away from the hydrophobic target binding pocket. For better visualization, we updated Figure 4D by reorienting the model and adding surface representation to emphasize the relative positions of binding pocket and cleavage sites. Furthermore, the thermal stability of CHP3 is not increased upon CBD binding in presence of Mg^2+^, whereas it increased in the presence of Ca^2+^ (figure 3, nanoDSF), while CBD binding increased CHP3 proteolytic stability in both conditions. We updated results as follows: “Both major cleavage sites are highly unlikely to be covered by CBD in the complex, as they are facing away from the target-binding pocket (Figure 4D). Thus, the drastic changes in the proteolytic stability indicate reduced flexibility in EF-2 and EF-4 which could be caused by structural rearrangements of CHP3 upon target peptide binding.

We agree that we did not include kinetic data and thus the term dynamics might be misleading. Thus, we omitted the term conformational dynamics and now consistently throughout the manuscript including Figure 7 refer to conformation (open or closed) or conformational changes. In addition, we use the term state in context of Mg^2+^- or Ca^2+^-bound state or apo-state.

The two-fold difference in affinity between CHP3 and mCHP3 for CBD is small, and thus one could argue that this is insignificant. However, it has a resemblance to some of the works on the disordered flanking regions of folded proteins that dynamically occupy a binding site in the folded domain (see e.g. works on RST and p53 with tails PMID: 34687712 and PMID: 30420502) – similar effect were seen. Would be a good addition to the discussion.

Thanks for bringing this up. We included in the discussion:

“As the effect on affinity is small, the myristoyl moiety binds most likely weakly or transiently to the target binding pocket. This resembles the dynamic regulatory interaction of disordered protein regions of p53 to its DNA-binding domain (65) and of Radical-Induced Cell Death 1 (RCD1) to the ligand-binding pocket (66).”

The suggested publications were added as references 65 and 66.

The discussion is lengthy and not forward-looking. The entire first paragraph is mostly a repetition of the introduction.

The discussion was updated as described in our answers to questions and comments. Further, the redundant part of the first and second paragraph was deleted.

It would have been appropriate to include other already mentions points above as well as a reflection of how the other calcium-binding sensitive partners to NHE1 (calmodulin and calcineurin) and the membrane interaction of NHE1 would jointly organize NHE1 at the membrane.

We included the following sentence in the discussion:

“This information would be valuable also to elucidate signal integration through the regulatory domain of NHE1, which acts as signaling hub interacting with several regulatory proteins including the ca^2+^-binding proteins calmodulin and calcineurin (3).”

In the discussion, it is concluded that Ca^2+^ induced higher flexibility. How is this conclusion reached? The same pattern of peptide cleavage is reached, and the protein is destabilized, which could be due to a different conformation as also probed with the hydrophobicity assay. It is not necessarily more flexible. Please elaborate on this to make the argumentation clearer or remove it from the discussion if not fully supported.

We updated the argumentation in results and discussion omitting the general term flexibility and now distinguishing between thermal stability (nanoDSF) and local flexibility (limited proteolysis). For updates in results, please see paragraph “Ca^2+^ binding to CHP3 allows more effective cleavage within EF-2, whereas CBD binding drastically increases proteolytic stability” and related updates in paragraph three of the discussion.

Also, in the discussion, it is suggested that the conformation of the N-terminal lobe is affected, which data supports this? How can this be inferred from the presented data? It is also indicated in Figure 7.

The argumentation is based on the effect of Ca^2+^ binding in the C-lobe on the limited trypsinolysis in the N-lobe. We clarified in the discussion as follows:

“Further, replacement of Mg^2+^ with Ca^2+^ at EF-3 (C-lobe) lowered the thermal stability of CHP3 and mCHP3 and enhanced proteolytic cleavage in EF-2 (N-lobe) suggesting an increase in local flexibility. Thus, Ca^2+^ induced conformational changes involve not only the C-lobe of CHP3 but also its N-lobe.”

The change in the amount of CHP3 in the liposome fraction upon CBD binding is very interesting and suggests that the myristoyl group and the CDB act cooperatively. Are there any relevant exposed residues on the exposed side of the bound CDB that can explain this? Lysine residues or hydrophobic residues? It would be good to have an AlphaFold2 structure of the complex to help address this.

As we had stated in the introduction:

“Membrane binding via the myristoyl moiety is usually enforced with either a second lipidation site or clusters of positively charged and/or hydrophobic residues (25, 26).”

We do not have such clusters in the CBD construct used. We also generated the model of CHP3 in complex with CBD using ColabFold (Mirdita et al. and Steinegger 2022, Nat. Methods, 19, 679-682, DOI: 10.1038/s41592-022-01488-1), which is based on AlphaFold2 (Ref. 42) and AlphaFold_multimer_v3. The model of the CHP3:CBD is very similar to published structures of CHP1 and CHP2 in complex with the target peptide. There is only one relevant residue at each end of the CBD peptide we used (residues 525-545), namely R525, and Tyr540, which might interact with a membrane. These single residues are unlikely to substantially contribute to the interaction. Noteworthy, there is a cluster of three lysine residues upstream of the CBD sequence (Lys518, Lys519, Lys520). These residues might further enhance the membrane association of myristoyl-anchored CHP3 in complex with NHE1. This is beyond the scope of the current study.

The 3 remaining EF-hands appear not to bind calcium, where are they mutated, and which properties do they then have? Please include.

EF-2 and EF-4 of CHP3 have 9-residue insertions within the Ca^2+^-coordinating loop, whereas EF-1 lacks conservative negatively charged residues important for Ca^2+^ coordination in positions X, Y and –Z. Thus, they still have an EF-hand-like fold, but are unable to bind Ca^2+^. We added this information to the legend of Figure 4—figure supplement 2.

FiguresFigure 1 – it appears for both proteins that a fraction of dimers elutes as well to the left of the main peak. Please comment.

CHP3 can form an intermolecular disulfide bond in the absence or even at a low concentration of reducing agent. This was previously noted in size exclusion chromatography experiments (Ref. 4). At the very C-terminus, the last but one residue is Cys213, which should be accessible for the reaction. We added to the figure legend of Figure 1, panel C:

“A shoulder with shorter arrival times indicates the presence of a low abundant dimeric species for CHP3 and mCHP3.”

At the same time, we did not observe CHP3 dimerization with analytical SEC (Figure 1—figure supplement 1) under reducing conditions.

Figure 4 – relevant to indicate the potential coverage (Ks and Rs in the sequence) and not just the two where you see cleavage. Aldo the additional bands on the mCHP3 should be acknowledged.

We added the potential coverage (K and R residues) of CHP3 in Figure 4—figure supplement 1 as panel A. We mentioned the presence of an additional band for mCHP3 as described in our answer to the question “For trypsinolysis, what is the coverage?”

Reviewer #3 (Recommendations for the authors):The description of the limited proteolysis experiment would benefit from additional detail. E.g., how many trypsin sites are in CBD? Is the stoichiometry of CBD:CHP3 1:1 or is an excess of CBD added? In the case of the latter, an experiment to determine whether CBD inhibits trypsin may be considered. Admittingly, this is unlikely to be the case, but it would further strengthen the claim of CBD-mediated stabilization.

The description of the limited proteolysis results was updated for clarity as described in response to reviewer #2 (see answer to question: “For trypsinolysis, what is the coverage?"). Further, there are no trypsin cleavage sites in CBD except the first residue, Arg525. We obtained the CHP3:CBD complex via co-expression in *E. coli* followed by co-purification. Due to formation of the stable complex, no free CBD is present in the reaction mix. The stoichiometry of CBD:CHP3 is 1:1, as previously shown. We updated the corresponding sentence in Introduction:

“We previously demonstrated that CHP3 binds at 1:1 ratio to the CHP-binding region of human NHE1 (CBD) with high affinity in the presence of Mg^2+^ (35).”